# EFFICIENT FAIRNESS-PERFORMANCE PARETO FRONT COMPUTATION

## ABSTRACT

There is a well known intrinsic trade-off between the fairness of a representation and the performance of classifiers derived from the representation. Due to the complexity of optimisation algorithms in most modern representation learning approaches, for a given method it may be non-trivial to decide whether the obtained fairness-performance curve of the method is optimal, i.e., whether it is close to the true Pareto front for these quantities for the underlying data distribution.

In this paper we propose a new method to compute the optimal Pareto front, which does not require the training of complex representation models. We show that optimal fair representations possess several useful structural properties, and that these properties enable a reduction of the computation of the Pareto Front to a compact discrete problem. We then also show that these compact approximating problems can be efficiently solved via off-the shelf concave-convex programming methods. Finally, in addition to representations, we show that the new methods may also be used to directly compute the Pareto front of fair classification problems.

Since our approach is independent of the specific model of representations, it may be used as the benchmark to which representation learning algorithms, or classifiers, may be compared. We experimentally evaluate the approach on a number of real world benchmark datasets.

## 1 INTRODUCTION

Fair representations are a central topic in the field of Fair Machine Learning, Mehrabi et al. (2021), Pessach & Shmueli (2022),Chouldechova & Roth (2018). Since their introduction in Zemel et al. (2013), Fair representations have been extensively studied, giving rise to a variety of approaches based on a wide range of modern machine learning methods, such GANs, variational auto encoders, numerous variants of Optimal Transport methods, and direct variational formulations. See the papers Feldman et al. (2015), Edwards & Storkey (2015),Louizos et al. (2015),Madras et al. (2018), Gordaliza et al. (2019); Zehlike et al. (2020), Song et al. (2019), Du et al. (2020), for a sample of existing methods.

For a given representation learning problem and a target classification problem, since the fairness constraints reduce the space of feasible classifiers, the best possible classification performance will usually be lower as the fairness constraint becomes stronger. This phenomenon is known as the Fairness-Performance trade-off. Assume that we have fixed a way to measure fairness. Then for a given representation learning method, one is often interested in the fairness-performance curve $(\gamma, E(\gamma))$. Here, $\gamma$ is the fairness level, and $E(\gamma)$ is the classification performance of the method at that level. The curve $(\gamma, E(\gamma))$ where $E(\gamma)$ is the best possible performance over all representations and classifiers under the constraint is known as the Fairness-Performance Pareto Front.

As indicated by the above discussion, representation learning methods typically involve models with high dimensional parameter spaces, and complex, possibly constrained non-convex optimisation algorithms. As such, these methods may be prone to local minima, and may be sensitive to a variety of hyper-parameters, such as architecture details, learning rates, and even initializations. While the representations produced by such methods may often be useful, it nevertheless may be difficult to decide whether their associated Fairness-Performance curve is close to the true Pareto Front.

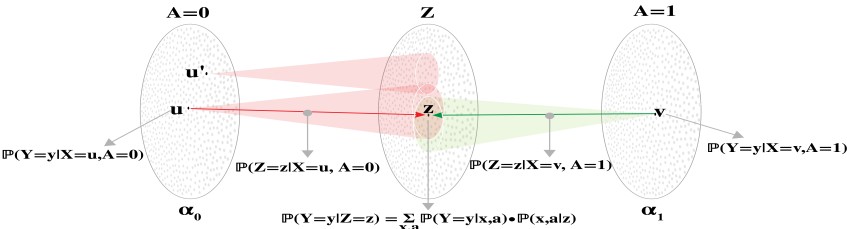

Figure 1: Illustration of fair representation learning setup, where representation space Z (center) connects input features X conditioned on protected attribute values A=0 (left) and A=1 (right). Multiple source points (x,a) can map to the same representation z, with each z maintaining target information through P(Y|Z=z) derived from the conditional probabilities P(Y|x,a) of its source points.

In this paper we propose a new method to compute the optimal Pareto front, which does not require the training of complex fair representation models. In other words, we show that, perhaps somewhat surprisingly, the computation of the Pareto Front can be decoupled from that of the representation, and only relies on learning of much simpler, unconstrained classifiers on the data. To achieve this, we first show that the optimal fair representations satisfy a number of structural properties. While these properties may be of independent interest, here we use them to express the points on the Pareto Front as the solutions of small discrete optimisation problems. The problems fall into the category of *concave minimisation* problems Benson (1995), which have been extensively studied, and can be efficiently solved using modern dedicated optimisation frameworks, Shen et al. (2016).

We now describe the results in more detail. Let $X$ and $A$ denote the data features and the sensitive attribute, respectively. The sensitive attribute is a part of the data, and we assume that $A$ is binary. The features $X$ will be taking values in $\mathbb{R}^d$. In addition, we have a *target* variable $Y$, taking values in a finite set $\mathcal{Y}$. We are then interested in representations that maximise the performance of prediction of $Y$, under the fairness constraint. The representation is denoted by $Z$, and is typically expressed by constructing the conditional distributions $\mathbb{P}(Z|X = x, A = a)$, for $a \in \{0, 1\}$. The problem setting is illustrated in Figure 1.

The remainder of the paper unfolds as follows: Algorithm 1 outlines the steps for computing the Pareto front - learning calibrated probability estimators, constructing relevant histograms, and solving the resulting optimization problem using standard solvers.
The theoretical foundation for this approach is built on two key results. The Factorization result (Section D) demonstrates that instead of working with all features x, we need only consider the distributions $P(Y|x)$, as inputs with identical distributions are equivalent for Pareto front purposes. This crucial insight reduces our representation space from $R^d$ to $\Delta Y$, making discretization more tractable. The Invertibility Theorem (Section 4) complements this by establishing bounds on the representation's target space - specifically, how many points z are needed in the representation to maintain accuracy.

As for a detailed breakdown of each section : The invertibility theorem states that in an *optimal* representation, every point $z$ can be generated from at most two $x$'s, one given $A = 0$ and another given $A = 1$. That is, there are *unique* $u, v$ such that $\mathbb{P}(Z = z|X = u, A = 0) \neq 0$ and $\mathbb{P}(Z = z|X = v, A = 1) \neq 0$. This in turn hints at a possibility that in an optimal representation, the space of $z$ may be indexed by the pairs $(u, v)$, where $u, v$ denote $x$'s corresponding to $A = 0$ and $A = 1$ as above. Indeed, an additional result, the Uniform Approximation Lemma D.1, deferred to Supplementary D due to space constraints, implies that for every pair $(u, v)$, one needs to have only a small number of $z$'s associated with it, in order to approximate all possible representations. Together, these results bound the number of points needed to model *all* optimal representations, if $X$ has finitely many values.

Next, in Section 5.1 we show that instead of discretising $X$ as suggested above, one may in fact discretise a much smaller space, i.e., $\Delta_{\mathcal{Y}}$—the space of all possible distributions of $Y$. As an example, if $Y$ is binary, this space is equivalent to the interval $[0, 1]$. This reduction is possible due to the *factorisation* property of representations. That is, we show that for the purposes of the computation

of the Pareto front, any representation may be represented as a composition (thus the factorisation terminology) of a mapping into $\Delta_{\mathcal{Y}}$, and a representation of $\Delta_{\mathcal{Y}}$.

Based on these results, in Section 5.2 we introduce MIFPO, a discrete optimisation problem for the computation of the Pareto front, and in Section 5.3 we provide details on estimating the problem parameters from the data. The MIFPO problem is a concave minimisation problem with linear constraints, and we solve it using the disciplined convex-concave programming framework, DCCP, Shen et al. (2016). It is worth noting that for the special case of perfect fairness, i.e., the point $\gamma = 0$ on the curve, the MIFPO problem is in fact a *linear programming* problem, solvable by standard means, and closely related to the Optimal Transportation problem (Supplementary B).

We illustrate MIFPO by evaluating it on a number of standard real world fairness benchmark datasets. We also compare its fairness-performance curve to that of FNF, Balunović et al. (2022a), a recent representation learning approach which yields representations with guaranteed fairness constraint satisfaction. As expected, MIFPO produces tighter fairness-performance curves, indicating that FNF still has space for improvement. This suggests that MIFPO may also be used as an analysis method in the construction of full representation learning algorithms. In addition, we show that MIFPO may also be used to evaluate the fairness-performance Pareto front of fair *classifiers* (Section 3), where fairness is measured by *statistical parity*. We thus evaluate two widely used fair classification methods and, similarly, find that their Pareto front is significantly below that of MIFPO.

To summarise, the contributions of this paper are as follows: **(a)** We derive several new structural properties of optimal fair representations. **(b)** We use these properties to construct a model independent problem, MIFPO, which can approximate the Pareto Front of arbitrary high dimensional data distributions, but is much simpler to solve than direct representation learning for such distributions. **(c)** We illustrate the approach on real world fairness benchmarks.

The rest of this paper is organised as follows: Section 2 discusses the literature. Section 3 introduces the formal problem setting. The Inveribility Theorem is discussed in Section 4, and the details of the MIFPO construction are given in Section 5. Experimental results are presented in Section 6, and we conclude the paper in Section 7. All proofs are provided in the Supplementary Material.

## 2 LITERATURE AND PRIOR WORK

As mentioned in the previous Section, there exists a variety of approaches to the construction of fair representations. We refer to the book Barocas et al. (2023), and surveys Mehrabi et al. (2021),Du et al. (2020), for an overview. Notably, the work Song et al. (2019) introduces a variational framework that unifies many of the existing methods under a single optimisation procedure. This work also explicitly deals with the difficulty of obtaining useful fairness-performance trade-offs, although such considerations are present in practically all fairness related work.

In a recent paper, Balunović et al. (2022a), it was observed that if one knows the density of the source data, $\mathbb{P}(X|A = a)$, $a \in \{0, 1\}$, and uses models with computable change of variables, such as the normalising flows, Rezende & Mohamed (2015), then one can verify the total variation fairness criterion (see (3)), with high probability. This enables one to provide guarantees on the fairness of a representation, which is clearly an important feature in applications. Such guarantees also allow result comparison between different methods. Indeed, if a given constraint is not guaranteed in one of the methods, then the performance comparison may be unfair. Due to this reason, in the experiments Section 6 we compare our own results to those of Balunović et al. (2022a). It may be worthwhile to note, however, that the assumption of knowing the densities $\mathbb{P}(X|A = a)$, made in Balunović et al. (2022a), is quite significant. This is due to the fact that such information is normally not given, and the estimation of densities from data is generally a difficult problem. We remark that our approach does not rely on this assumption.

Regarding related work dealing with optimal tradeoffs between accuracy and fairness, in Zhao & Gordon (2019) the autors developed a theoretical error bound for the *perfecly fair* classifier and are not concerned with the full Pareto front, nor addressing fair representation. Dehdashtian et al. (2024),Sadeghi et al. (2022) proposed learning fair representations using neural encoders with RKHS mappings and classifiers, measuring independence-based fairness through RKHS proxies. This approach requires implicit data assumptions, particularly Gaussian projections in Sadeghi et al. (2022) in contrast to our method which also provide simple solution. The most related work Wang

et al. (2024) builds on Kim et al. (2020) observation that the classification Pareto front depends solely on classifier confusion matrices. While similar to our factorization result (Section 5.1), their main contribution characterizes these matrices through infinite constraint intersection, proposing an iterative algorithm that progressively adds constraints to solve increasingly constrained Pareto problems. Like our approach, their method learns a probability estimator $g(x, a) = P(y|x, a)$ for constraints. However, we utilize g differently - leveraging optimal fair representation properties to formulate a closed-form problem with finite constraints solvable by standard methods, rather than requiring custom iterative algorithms. Moreover, our approach addresses the broader challenge of representation learning, not just classification. While Wang et al. (2024) handles multi-class classification and non-binary discrete attributes, their iterative algorithm operates in high-dimensional spaces with potentially exponential constraint growth, compared to our solution's required simplex $\Delta Y$ discretization, especially when implemented thoughtfully using clustering algorithms.

## 3 PROBLEM SETTING

Let $A$ be a binary sensitive variable, and let $X$ be an additional feature random variable, with values in $\mathbb{R}^d$. Assume also that there is a target variable $Y$ with finitely many values, jointly distributed with $X, A$.

A representation $Z$ of $(X, A)$ is defined as a random variable taking values in some space $\mathbb{R}^{d'}$, with *(i)* distribution given through $\mathbb{P}_\theta(Z|X, A)$, where $\theta$ are the parameters of the representation, and *(ii)* such that $Z$ is independent of the rest of the variables of the problem conditioned on $(X, A)$. In particular, we have

$$Z \perp\!\!\!\perp Y|(X, A), \tag{1}$$

where $\perp\!\!\!\perp$ denotes statistical independence.

Fairness in this paper will be measured by the Total Variation distance. For two distributions, $\mu, \nu$ on $\mathbb{R}^d$, with densities $f_\mu, f_\nu$, respectively, this distance is defined as

$$\|\mu - \nu\|_{TV} = \frac{1}{2} \sup_{g \text{ s.t. } \|g\|_\infty \leq 1} \int g(x) \cdot [f_\mu(x) - f_\nu(x)] \, dx = \frac{1}{2} \int |f_\mu(x) - f_\nu(x)| \, dx. \tag{2}$$

Note that $\int |f_\mu(x) - f_\nu(x)| \, dx$ is in fact the $L_1$ distance, and the equivalence $\|\cdot\|_{TV} = \frac{1}{2} \|\cdot\|_{L_1}$ is well known, see Cover & Thomas (2012).

For $a \in \{0, 1\}$, let $\mu_a$ be the distribution of $Z$ given $A = a$, i.e. $\mu_a(\cdot) := \mathbb{P}(Z = \cdot|A = a)$. For $\gamma \geq 0$, we say that the representation $Z$ is $\gamma$-fair iff

$$\|\mu_0 - \mu_1\|_{TV} \leq \gamma \qquad \text{(Fairness Condition)}. \tag{3}$$

In what follows we will denote the norm of the difference as $F_\theta := \|\mu_0 - \mu_1\|_{TV}$. Note that (3) is a quantitative relaxation of the "perfect fairness" condition in the sense of statistical parity, which requires $Z \perp\!\!\!\perp A$. Specifically, observe that by definition, $Z \perp\!\!\!\perp A$ iff (3) holds with $\gamma = 0$ (i.e. $\mu_0 = \mu_1$). In addition, as shown in Madras et al. (2018), (3) implies several other common fairness criteria. In particular, (3) implies bounds on demographic parity and equalized odds metrics for any downstream classifier built on top of $Z$.

Next, we describe the measurement of information loss in $Y$ due to the representation. Let $h : \Delta_{\mathcal{Y}} \to \mathbb{R}$ be a continuous and concave function on the set of probability distributions on $\mathcal{Y}$, $\Delta_{\mathcal{Y}}$. The quantity $h(\mathbb{P}(Y|X = x))$ will measure the best possible prediction accuracy of $Y$ conditioned on $X = x$, for varying $x$. As an example, consider the case of binary $Y$, $\mathcal{Y} = \{0, 1\}$. Every point in $\Delta_{\mathcal{Y}}$ can be written as $(p, 1 - p)$ for $p \in [0, 1]$, and we may choose $h$ to be the optimal binary classification error,

$$h((1 - p, p)) = min(p, 1 - p). \tag{4}$$

Another possibility it to use the entropy, $h((1 - p, p)) = p \log p + (1 - p)log(1 - p)$. The average uncertainty of $Y$ is given by $\mathbb{E}_{x \sim X} h(\mathbb{P}(Y|X = x))$. Note that this notion does not depend on a particular classifier, but reflects the performance the *best* classifier can possibly achieve (under appropriate cost).

The goal of fair representation learning is then to find representations $Z$ that for a given $\gamma \geq 0$ satisfy the constraint (3), and under that constraint minimize the objective $E = E_\theta$ given by

$$E_\theta = \mathbb{E}_{z \sim Z} h(\mathbb{P}(Y|Z = z)). \tag{5}$$

That is, the representation should minimise the optimal $Y$ prediction error (using $Z$) under the fairness constraint.

The curve that associates to every $0 \leq \gamma \leq 1$ the minimum of (5) over all representations $Z$ which satisfy (3) with $\gamma$ is referred to as the *Pareto Front* of the Fairness-Performance trade-off.

In supplementary material Section A we show that for any representation, $\mathbb{E}_{z \sim Z} h(\mathbb{P}(Y|Z = z)) \geq \mathbb{E}_{x \sim X} h(\mathbb{P}(Y|X = x))$, i.e. representations generally decrease the performance. Technically, this happens due to the concavity of $h$, and this result is an extension of the classical *information processing inequality*, Cover & Thomas (2012). To intuitively see *why* representations decrease performance, note that as illustrated in Figure 1, the distribution $\mathbb{P}(Y|Z = z)$ is a mixture of various distributions $\mathbb{P}(Y|X = x, A = a)$. Thus, for instance, even if all $\mathbb{P}(Y|X = x, A = a)$ are deterministic, but not all of the same value, $\mathbb{P}(Y|Z = z)$ will not be deterministic.

We conclude this section by showing that the Pareto front of binary classifiers, with statistical parity fairness measure, can be computed from the Pareto front of representations with total variation fairness measure. In fact, Lemma 3.1 below states that both Pareto fronts amount to the same curve. As discussed in Section 1, this equivalence implies that MIFPO can be used to evaluate fair classifiers, in addition to fair representations.

Consider a binary classifier $\hat{Y}$ of $Y$, with $(X, A)$ as features. The prediction error is defined as usual by $\epsilon(\hat{Y}) := \mathbb{P}\left(\hat{Y} \neq Y\right)$. The statistical parity *distance* of $\hat{Y}$ is defined as

$$\Delta_{SP}(\hat{Y}) := \left| \mathbb{P}\left(\hat{Y} = 1|A = 1\right) - \mathbb{P}\left(\hat{Y} = 1|A = 0\right) \right|. \tag{6}$$

Let the uncertainy measure $h$ be defined by (4).

**Lemma 3.1.** *Let $\hat{Y}$ be a classifier of $Y$. Then there is a representation given by a random variable $Z$ on a set $\mathcal{Z}$ with $|\mathcal{Z}| = 2$, such that*

$$\mathbb{E}_{z \sim Z} h(\mathbb{P}(Y|Z = z)) \leq \epsilon(\hat{Y}) \text{ and } \|\mu_0 - \mu_1\|_{TV} \leq \Delta_{SP}(\hat{Y}). \tag{7}$$

*Conversely, for any given representation $Z$, there is a classifier $\hat{Y}$ of $Y$ as a function of $Z$ (and thus of $(X, A)$), such that*

$$\epsilon(\hat{Y}) \leq \mathbb{E}_{z \sim Z} h(\mathbb{P}(Y|Z = z)) \text{ and } \Delta_{SP}(\hat{Y}) \leq \|\mu_0 - \mu_1\|_{TV}. \tag{8}$$

The Proof of Lemma 3.1 is presented in Supplementary Material Section H.

## 4 THE INVERTIBILITY THEOREM

In this section we define the notion of invertibility for representations, and show that optimal fair representations are invertible.

To this end, let us first introduce an additional notation, which is more convenient when describing representations specifically on finite sets. Let $S_0, S_1$ be finite disjoint sets, where $S_a$ represents the values of $(X, A)$ when $A = a$, for $a \in \{0, 1\}$. Denote $S = S_0 \cup S_1$. We are assuming that there is a probability distribution $\zeta$ on $S$, and $A$ is the random variable $A = \mathbb{1}_{\{s \in S_1\}}$. $X$ is defined as taking the values $s \in S$, with $\mathbb{P}(X = s) = \zeta(s)$. Further, the variable $Y$ is defined to take values in a finite set $\mathcal{Y}$, and for every $s \in S$, its conditional distribution is given by $\rho_s \in \Delta_{\mathcal{Y}}$. That is, $\mathbb{P}(Y = y|X = s, A = a) = \rho_s(y) = \rho_{s,a}(y)$. [1] This completes the description of the data model.

For $a \in \{0, 1\}$ we denote $\alpha_a = \mathbb{P}(A = a) = \zeta(S_a)$, and $\beta_a(s) = \mathbb{P}(X = s|A = a)$. Observe that $\beta_a(s) = 0$ if $s \notin S_a$, and $\beta_a(s) = \zeta(s)/\zeta(S_a)$ if $s \in S_a$.

We now describe the representation. The representation will take values in a finite set $\mathcal{Z}$. For every $s \in S$ and $z \in \mathcal{Z}$, let $T_a(z, s) = \mathbb{P}(Z = z|X = s, A = a)$ be the conditional probability of

---

[1]Note that there is a slight redundancy in the notation $\mathbb{P}(Y = y|X = s, A = a)$ here, since $a$ is determined by $s$. However, to retain compatibility with the standard notaion, we specify them both. This is similar to the continuous situation, in which although $A$ is technically part of the features, $X$ and $A$ are specified separately.

representing $s$ as $z$. Then $T_a$'s fully define the distribution of the representation $Z$. We shall refer to the representation as $T$ or $Z$ in interchangeably. Finally, for $a \in \{0, 1\}$ denote

$$\mu_a(z) = T_a \beta_a = \mathbb{P}\left(Z = z | A = a\right) = \sum_{s \in S_a} \beta_a(s) T_a(z, s). \tag{9}$$

Similarly to the previous section, our goal is to find representations $T$ that minimise the cost (5) under the constraint (3), for some fixed $\gamma > 0$.

A representation $T$ is *invertible* if for every $z \in \mathcal{Z}$ and every $a \in \{0, 1\}$, there is at most one $s \in S_a$ such that $T_a(z, s) > 0$. In words, a representation is invertible, if any given $z$ can be produced by at most two original features $s$, and at most one in each of $S_0$ and $S_1$.

Given a representation $T$ and $z \in \mathcal{Z}$, we say that an $s \in S$ is a *parent* of $z$ if $T_a(z, s) > 0$ for the appropriate $a$.

**Theorem 4.1.** *Let $T$ be any representation of $(S, A, \zeta, Y)$ on a set $\mathcal{Z}$. Then there exists an invertible representation $T'$ of $(S, A, \zeta, Y)$, on some set $\mathcal{Z}'$, such that*

$$\mathbb{E}_{z' \sim Z'} h(\mathbb{P}\left(Y | Z' = z'\right)) \leq \mathbb{E}_{z \sim Z} h(\mathbb{P}\left(Y | Z = z\right)) \text{ and } \|\mu'_0 - \mu'_1\|_{TV} = \|\mu_0 - \mu_1\|_{TV}. \tag{10}$$

Here, similarly to $\mu_a$, $\mu'_a$ are the distributions induced by $T'$ on $\mathcal{Z}'$. In words, for every representation, we can find an invertible representation of the same data which satisfies at least as good a fairness constraint, and has at least as good performance as the original. In particular, this implies that when one searches for optimal performance representations, it suffices to only search among the invertible ones.

The proof proceeds by observing that if an atom $z \in \mathcal{Z}$ has more than one parent on the same side (i.e. $S_0$ or $S_1$), then one can split this atom into two, with each having less parents. However, the details of this construction are somewhat intricate. The full argument can be found in Section C.

## 5 THE MODEL INDEPENDENT OPTIMIZATION PROBLEM

This Section describes the specific numerical problem, MIFPO, that is used to compute the Fairness-Performance Pareto Front. Section 5.1 discusses the factorisation of representations, in Section 5.2 the MIFPO problem is defined, and in Section 5.3 we describe how MIFPO is instantiated from data.

To motivate MIFPO, let us write again some of the terms involved in the expressions of the performance cost and the fairness constraints $E$ and $F$, as defined in Section 3. To simplify the exposition, we use the discrete summation notation, but similar expressions can also be written using integration and densities. We have

$$\mathbb{P}(z) = \sum_{x,a} \mathbb{P}(z|x,a) \mathbb{P}(x,a) \text{ and } \mathbb{P}(z|a) = \sum_{x} \mathbb{P}(z|a,x) \mathbb{P}(x,a) / \mathbb{P}(a). \tag{11}$$

Moreover,

$$\mathbb{P}(Y|z) = \sum_{x,a} \mathbb{P}(Y|x,a,z) \mathbb{P}(a,x|z) = \sum_{x,a} \mathbb{P}(Y|x,a) \mathbb{P}(z|a,x) \mathbb{P}(a,x) / \mathbb{P}(z), \tag{12}$$

where we have used $\mathbb{P}(Y|x,a,z) = \mathbb{P}(Y|x,a)$, which holds due to the independence condition (1) of the representation. We observe that one can write the performance cost and the fairness constraint, $E$ and $F$, in terms of $\mathbb{P}(a)$, $\mathbb{P}(x,a)$, $\mathbb{P}(Y|x,a)$, and of the representation $\mathbb{P}(z|x,a)$. We refer to these quantities as the *basic probabilities*. It follows that *any two problems having the same basic probabilities, will have the same $E$ and $F$.*

One way to view the construction of the MIPFO problem is simply as a discretization, where we model the above basic probabilities in an abstract way, on a finite set, without the need to consider the details of how the representation $\mathbb{P}(z|x,a)$ is implemented. However, discretization of the features $x \in \mathbb{R}^d$ directly would likely always be too crude. In the following Section we show that in fact one only needs to discretize a much smaller space, namely $\Delta_{\mathcal{Y}}$ - the set of distributions over $Y$ values, which is typically easy to achieve. In Section 5.3 we show how factorisation is used in practice.

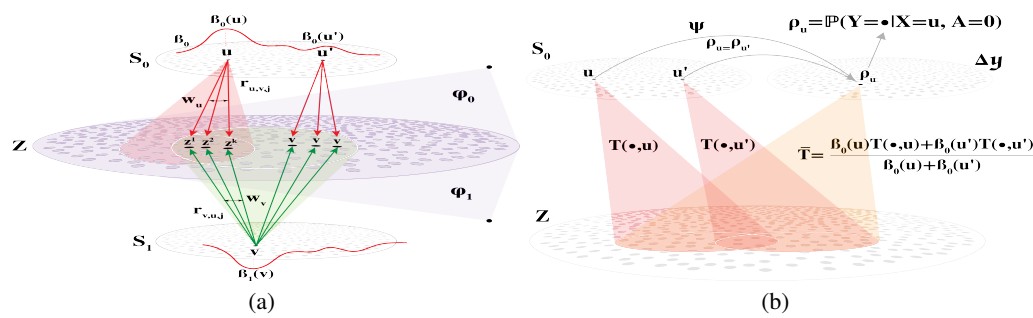

Figure 2: (a) The MIFPO Setting (b) Factorisation Through $\Delta_{\mathcal{Y}}$

## 5.1 FACTORISATION

The general idea behind factorisation is the observation that in a given representation, individual source points $x, x'$ may be merged if they have the same $\rho_x = \mathbb{P}(Y|x)$ distribution, without affecting the measurements $E$ and $F$. We now carry out this argument formally, with Figure 2(b) illustrating the construction. For clarity, we use the notation of Section 4 for finite sets. The extension to the continuous case is discussed in the end of the Section.

Suppose that we have a representation on some, possibly very large source space $S = S_0 \cup S_1$, with representation space $\mathcal{Z}$. Introduce a map $\Psi : S_0 \to \Delta_{\mathcal{Y}}$, given by $\Psi(v) = \rho_v$. Then we can define a distribution $\bar{\beta}_0$ on $\Delta_{\mathcal{Y}}$, and the transition kernel $\bar{T}_0 : \mathcal{Z} \times \Delta_{\mathcal{Y}} \to [0, 1]$ by

$$\bar{\beta}_0(\rho) = \sum_{v \in \Psi^{-1}(\{\rho\})} \beta_0(v) \quad \text{and} \quad \bar{T}_0(z, \rho) = \sum_{v \in \Psi^{-1}(\{\rho\})} \frac{\beta_0(v)}{\bar{\beta}_0(\rho)} T_0(z, v), \text{ for all } \rho \in \Delta_{\mathcal{Y}}. \quad (13)$$

Similar construction can be made for $a = 1$, defining $\bar{\beta}_1, \bar{T}_1$. Observe that together, $\bar{\beta}_a$ and $\bar{T}_a$ define a representation on $\Delta_{\mathcal{Y}}$ with values in $\mathcal{Z}$ (technically, the source space is $\Delta_{\mathcal{Y}}^0 \cup \Delta_{\mathcal{Y}}^1$, where $\Delta_{\mathcal{Y}}^a$ are two disjoint copies of $\Delta_{\mathcal{Y}}$). Note that it is understood here that the distribution of $Y$ at point $\rho \in \Delta_{\mathcal{Y}}$ is taken to be $\rho$ itself. The crucial property of this new representation is that the induced distributions on $\mathcal{Z}$ coincide with those of the old one. That is,

$$\mu_a(z) = \bar{\mu}_a(z) \quad \text{and} \quad \mathbb{P}_T(Y|Z = z) = \mathbb{P}_{\bar{T}}(Y|Z = z) \quad (14)$$

for all $z \in \mathcal{Z}$. This property follows simply by substituting the definitions, although the derivation is slightly notationally heavy. The full details may be found in Supplementary Section F.2.

The meaning of (14) is that from $\mathcal{Z}$ one can not distinguish between the representations $T$ and $\bar{T}$, and in particular they have the same $E$ and $F$. We say that $T$ is *factorised* through $\Delta_{\mathcal{Y}}$ since it can be represented as a composition, of first applying $\Psi$ and then applying $\bar{T}$.

When the feature space $S$ is continuous, e.x. $\mathbb{R}^d$, the construction of $\bar{T}$ is similar. Indeed, $\Psi$ can be defined in the same way, $\bar{\beta}_a$ are defined as push-forwards $\bar{\beta}_a = \Psi \beta_a$, and $\bar{T}_a(z, \rho)$ is defined as conditional expectation, $\bar{T}_a(z, \rho) = \mathbb{E}_{v \sim \beta_0} T_a(z, v)|(\Psi(v) = \rho)$.

## 5.2 MIFPO DEFINITION

Fix an integer $k \geq 2$. The representation space $\mathcal{Z}$ will be a finite set which can be written as $\mathcal{Z} = S_0 \times S_1 \times [k]$, where $[k] := \{1, 2, \dots, k\}$. That is, every point $z \in \mathcal{Z}$ corresponds to some triplet $(u, v, j)$, with $u \in S_0, v \in S_1, j \in [k]$. The motivation for this choice of $\mathcal{Z}$ is as follows: First, note the by the invertibility Theorem 4.1, in an optimal representation, every $z \in \mathcal{Z}$ is associated with a pair of parents, $(u, v)$. Moreover, it can be shown that for every pair $(u, v)$, it is sufficient to have at most $k$ points $z$ which have $(u, v)$ as parents, to approximate *all* representations on *all* possible spaces $\mathcal{Z}'$, in terms of their $E$ and $F$ quantities, to a given degree $\varepsilon$. Here, $k$ would depend on $\varepsilon$ and on the function $h$, but not on the particular representations, or even the sizes $|S_0|, |S_1|$, etc. See Section D in Supplementary Material, for additional details. As a result, choosing $\mathcal{Z} = S_0 \times S_1 \times [k]$ should be sufficient to model all representations well. We have used $k = 10$ in all experiments.

The variables of the problem, modeling the representation itself, will be denoted by $r_{u,v,j}$ and $r_{v,u,j}$, for $(u,v,j) \in \mathcal{Z}$. These values correspond to $r_{u,v,j} = \mathbb{P}\left(Z = (u,v,j)|(X,A) = (u,0)\right)$ and $r_{v,u,j} = \mathbb{P}\left(Z = (u,v,j)|(X,A) = (v,1)\right)$. Note that the order of $u, v$ in $r_{u,v,j}$ and $r_{v,u,j}$ determines whether the weight is transferred from $S_0$ or $S_1$. The values $r$ correspond to those of $T_0(z,u), T_1(z,v)$ in Section 4. However, due to the special structure of the representation here, many values will be 0, such as for instance $T_0((u,v,j), u')$, whenever $u \neq u'$. The $r$ notation accounts for this, and is thus clearer. The full setup is illustrated in Figure 2(a).

Note that the variables represent probabilities, and thus satisfy the following constraints:

$$r_{u,v,j} \geq 0, \quad r_{v,u,j} \geq 0 \quad \forall (u,v,j) \in \mathcal{Z} \tag{15}$$

$$\sum_{v \in S_1, j \in [k]} r_{u,v,j} = 1 \quad \forall u \in S_0 \text{ and } \sum_{u \in S_0, j \in [k]} r_{v,u,j} = 1 \quad \forall v \in S_1. \tag{16}$$

Recall from Section 4 that the additional parameters of the problem are the conditional distributions $\beta_0(u), \beta_1(v)$, on $S_0$ and $S_1$, modeling $\mathbb{P}\left((X,A)|A = 0\right)$ and $\mathbb{P}\left((X,A)|A = 1\right)$ respectively. The quantities $\alpha_a = \mathbb{P}\left(A = a\right)$, and the conditional $Y$ distributions, $\rho_u, \rho_v \in \Delta_{\mathcal{Y}}$, modelling $\rho_u = \mathbb{P}\left(Y = \cdot|(X,A) = (u,0)\right), \rho_v = \mathbb{P}\left(Y = \cdot|(X,A) = (v,1)\right)$. Here we assume that these quantities are given, and in Section 5.3 below we describe how one may estimate them from the data.

With these preparations, we are ready to write the performance cost (5) in the new notation:

$$E_r = \sum_{z=(u,v,j)} \left(\alpha_0 \beta_0(u) r_{u,v,j} + \alpha_1 \beta_1(v) r_{v,u,j}\right) h \left(\frac{\rho_u \alpha_0 \beta_0(u) r_{u,v,j} + \rho_v \alpha_1 \beta_1(v) r_{v,u,j}}{\alpha_0 \beta_0(u) r_{u,v,j} + \alpha_1 \beta_1(v) r_{v,u,j}}\right). \tag{17}$$

Indeed, observe that due to the structure of our representations, every $z$ has two parents, and we have $\mathbb{P}\left(Z = z\right) = (\alpha_0 \beta_0(u) r_{u,v,j} + \alpha_1 \beta_1(v) r_{v,u,j})$. Similarly, $\mathbb{P}\left(Y|Z = z\right)$ is computed via (12) and substituted inside $h$ to obtain (17). As we show in Supplementary E, the cost (17) is a *concave* function of the parameters $r$.

We now proceed to discuss the fairness constraint $F$. Recall that we define $\mu_a(z) = \mathbb{P}\left(Z = z|A = a\right)$, for $a \in \{0, 1\}$. For $z = (u,v,j)$ we have then $\mu_0((u,v,j)) = \beta_0(u) r_{u,v,j}$ and $\mu_1((u,v,j)) = \beta_1(v) r_{v,u,j}$. We thus can write

$$F_r = \|\mu_0 - \mu_1\|_{TV} = \frac{1}{2} \sum_z |\mu_0(z) - \mu_1(z)| = \frac{1}{2} \sum_{(u,v,j)} |\beta_0(u) r_{u,v,j} - \beta_1(v) r_{v,u,j}| \tag{18}$$

and the Fairness constraint, for a given $\gamma \in [0,1]$ is simply

$$F_r \leq \gamma. \tag{19}$$

Although the constraint (19) is convex in the variables $r$, and can be incorporated directly into most optimisation frameworks, it may be significantly more convenient to work with *equality* constraints. Using the following Lemma, we can find equivalent equality constraints in a particularly simple form.

**Lemma 5.1.** *Let $\mu_0, \mu_1 \in \Delta_{\mathcal{Z}}$ be two probability distributions over $\mathcal{Z}$ and fix some $\gamma \geq 0$. If $\|\mu_0 - \mu_1\|_{TV} = \gamma$ then there exist $\phi_0, \phi_1 \in \Delta_{\mathcal{Z}}$ such that $\mu_0 + \gamma\phi_0 = \mu_1 + \gamma\phi_1$. In the other direction, if there exist $\phi_0, \phi_1 \in \Delta_{\mathcal{Z}}$ such that $\mu_0 + \gamma\phi_0 = \mu_1 + \gamma\phi_1$, then $\|\mu_0 - \mu_1\|_{TV} \leq \gamma$.*

As consequence, if we find distributions $\phi_0, \phi_1 \in \Delta_{\mathcal{Z}}$ such that $\mu_0 + \gamma\phi_0 = \mu_1 + \gamma\phi_1$ holds, then we know that (19) also holds, and conversely, if (19) holds, then distributions as above exist.

Using this observation, we introduce new variables, $\phi_{u,v,j}$ and $\phi_{v,u,z}$, for every $(u,v,j) \in \mathcal{Z}$, which correspond to $\phi_0((u,v,j))$ and $\phi_1((u,v,j))$ respectively. These variables will be required to satisfy the following constraints:

$$r_{u,v,j} \geq 0, \quad r_{v,u,j} \geq 0 \quad \forall (u,v,j) \in \mathcal{Z} \tag{20}$$

$$\sum_{u,v,j} r_{u,v,j} = 1 \text{ and } \sum_{u,v,j} r_{v,u,j} = 1 \tag{21}$$

$$\beta_0(u) r_{u,v,j} + \gamma\phi_{u,v,j} = \beta_1(v) r_{v,u,j} + \gamma\phi_{v,u,j} \quad \forall (u,v,j) \in \mathcal{Z}. \tag{22}$$

Here the first two lines encode the fact that $\phi_0, \phi_1$ are probabilities, while the third line encodes the fairness constraint, as discussed above. We now summarise the full MIFPO problem:

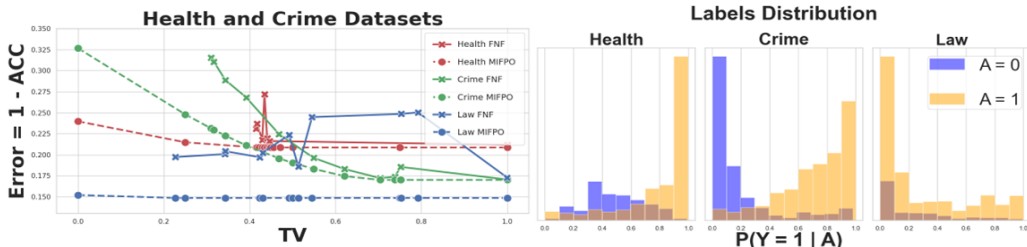

Figure 3: (a) Fairness-Performance Curves of MIFPO and FNF (b) Histograms $H_a$ for the Datasets

**Definition 5.1** (MIFPO). *For a fixed finite ground set $S = S_0 \cup S_1$, the problem parameters are the weight $\alpha_0$, the distributions $\beta_0, \beta_1$, on $S_0$ and $S_1$ respectively, and the distributions $\rho_x \in \Delta_{\mathcal{Y}}$ for every $x \in S$. The problem variables are $\{r_{u,v,j}, r_{v,u,j}, \phi_{u,v,j}, \phi_{v,u,j}\}_{(u,v,j) \in \mathcal{Z}}$ as defined above. We are interested in minimising the concave function* (17), *subject to the constraints* (15)-(16) *and* (20)-(22).

The relationship between MIFPO and the Optimal Transport problem is detailed in Section B.

## 5.3 Constructing a MIFPO Instance

In this Section we discuss the estimation of the quantities $\alpha, \beta$ and $\rho$ from the data, and specify how the data is used to construct the MIFPO problem instance.

For clarity we treat the case of binary $Y$, which is also the case for all datasets in Balunović et al. (2022a), on which we base our experiments. Note that for binary $Y$, $\Delta_{\mathcal{Y}}$ is simply the interval $[0, 1]$, with $p \in [0, 1]$ corresponding to the probability $p = \mathbb{P}(Y = 1|x, a)$.

The computation proceeds in 3 steps, schematically shown as Algorithm 1. The input is a labeled dataset $\{(x_i, a_i, y_i)\}_{i \leq N}$, with $x_i \in \mathbb{R}^d, a_i, y_i \in \{0, 1\}$, and two integers, $L, k$, which will specify the approximation degrees for the discretisation and for the approximation space, respectively. In Step **1.**, separately for the parts of the data corresponding to $A = 0$ and $A = 1$, we learn *calibrated* classifiers $c_0, c_1 : \mathbb{R}^d \to [0, 1]$, which represent the probabilities $c_a(x) = \mathbb{P}(Y = 1|X = x, A = a) = \rho_a(x)$. Note the following points: **(a)** In most applications, the true values of $\mathbb{P}(Y = 1|X = x, A = a)$ are non-binary (i.e. $\neq 0, 1$), and thus the $\{0, 1\}$-valued labels are only noisy observations of the actual probabilities. Thus, some sort of estimator must be used to learn the probabilities. **(b)** Typical classification objectives tend to yield in an imprecise estimation of $\mathbb{P}(Y = 1|X = x, A = a)$. However, this can usually be fixed by the process of calibration in post-processing, see Niculescu-Mizil & Caruana (2005), Kumar et al. (2019), (Berta et al., 2024). In Step **2.** we define the sets $H_a = \{c_a(x) \mid x \in X_a\} \subset [0, 1]$, and approximate them by histograms with $L$ bins. That is, for each $a$, we find $L$ points $\rho_a^1, \ldots, \rho_a^L \in [0, 1]$, and weights $\beta_a^1, \ldots, \beta_a^L$, such that the distribution with centers $\rho_a^i$ and weights $\beta_a^i$ approximates $H_a$ well. We typically take uniform bins, i.e. simply $\rho_a^l = \frac{l-1}{L-1}$, for $l = 1, \ldots, L$. See Figure 3(b) for an example of such histograms on real data. Finally, in Step **3.** we solve the MIFPO problem, as given in Definition 5.1. The ground set here is $S = S_0 \cup S_1$, with $S_0 = \{1, \ldots, L\}$ and $S_1 = \{1', \ldots, L'\}$, where we have added $'$ in $S_1$ to signify that the sets are disjoint copies in $S$, and $|S| = 2L$. The problem parameters are $\alpha_a^l, \beta_a^l, \rho_a^l$, as computed in the previous step, and the parameter $k$ determining the size of $\mathcal{Z}$.

## 6 Experiments

This section describes the evaluation of the MIFPO approach on real data. As indicated in Algorithm 1, two main components are required to apply our methods: building a calibrated classifier and solving the discrete optimization problem described in Section 5.2. For the calibrated classifier, we have used a standard XGBoost (Chen et al., 2015), with Isotonic Regression calibration, Pedregosa et al. (2011). Next, as discussed in Sections 1, 5.2, MIFPO is a concave minimisation problem, under linear constraints. While such problems do not necessarily posses the favourable properties of concave *maximisation* (or convex minimisation), such as unique minima points, it is nevertheless possible to

take advantage of concavity, to improve on general purpose optimisation methods. Here we have used the DCCP framework and the associated solver, (Shen et al., 2016; 2024). This framework is based on the combination of convex-concave programming ideas (CCP) Lipp & Boyd (2016) and disciplined convex programming, Grant et al. (2006). Throughout the experiments, we use the missclassification error loss $h$ given by (4). Supplementary Section G provides additional implementation details.

We evaluated MIFPO and the FNF fair representation learning algorithm, Balunović et al. (2022a), on the "Health", "Crime", and "Law" datasets, which are standard benchmarks in fairness-related work, and were also used in Balunović et al. (2022a). Figure 3(a) presents the MIFPO results as dotted lines, while the FNF results are solid lines. Figure 3(b) shows the histograms $H_a$ for the datasets (see the related discussion in Section 5.3).

First, as expected, when the histograms of $A = 0$ and $A = 1$ classes look similar, as in "Law" (indicating approximately $Y \perp\!\!\!\perp A$, fair $Y$), the Pareto front is flat. Otherwise, the front exhibits variability, as in "Health" and "Crime".

---

**Algorithm 1** MIFPO Implementation

**Input:** Data $\{(x_i, a_i, y_i)\}_{i \leq N}$, integers $L, k$.
  For $a \in \{0, 1\}$ denote $X_a = \{x_i \mid a_i = a\}$.
**1.** Learn *calibrated* classifiers
  $c_0, c_1 : \mathbb{R}^d \to [0, 1]$, such that
  $c_a(x) \sim \mathbb{P}(Y = 1 | X = x, A = a)$
**2.** Construct the histograms $\{\rho_a^l, \beta_a^l\}_{l=1}^L$,
  $a \in \{0, 1\}$, for the sets
  $H_a = \{c_a(x) \mid x \in X_a\} \subset [0, 1]$.
**3.** Solve MIFPO, given by Definition 5.1,
  with parameters $k$ and
  $\{\rho_a^l, \beta_a^l\}_{l=1}^L$ and $\alpha_a = |\{i \mid a_i = a\}| / N$.

---

Then, inspecting Figure 3(a), we observe that FNF struggles to obtain the full range of total variation values $\gamma$ in its fairness-performance curve. Moreover, for the values that are obtained, the accuracy computed by MIFPO at the corresponding points $\gamma$ is better. This highlights the difficulties with the construction of fair representations, which were circumvented in this paper.

As shown in Section 3, MIFPO may also be used to compute the Pareto Front for the fair classification problem. In the Supplementary Material Section H We evaluate the performance of MIFPO in compare to fair classifiers. These evaluations demonstrate that standard fair classifiers achieve a considerably lower accuracy than what is theoretically possible, and, the existing methods are also unable to present solutions for the full range of the statistical parity values.

Our approach is subject to three types of uncertainty: (i) Learning probability estimators from finite data - an unavoidable challenge in any ML approach. (ii) Discretization errors from histogram binning and representation space sizing. These are controllable: discretizing simple spaces like [0,1] is well-understood, and we provide bounds relating k to maximum front error (Supplementary Material Section D). (iii) DCCP potentially yielding sub-optimal results. While DCCP typically performs well, branch-and-bound methods can guarantee global optimum convergence if needed, though potentially at higher computational cost. This optimization domain is generally well-developed.

## 7 CONCLUSIONS, LIMITATIONS, AND FUTURE WORK

In this paper we have uncovered new fundamental properties of optimal fair representations, and used these properties to develop a model independent procedure for the computation of Fairness-Performance Pareto front from data. We have also demonstrated the procedure on real datasets, and have shown that it may be used as a benchmark for representation learning algorithms.

We now discuss limitations and a few possible directions for future work. Perhaps the biggest limitation of the current approach is the need to discretise the $\Delta_{\mathcal{Y}}$. This means that the $Y$ can only take a small number of values, which, for instance, prevents us from considering multiple target variables, i.e. $Y = (Y_1, \ldots, Y_t)$. It is thus of interest to ask whether it is possible to obtain a continuous version of our method, which would not discretise $\Delta_{\mathcal{Y}}$. Note that this would still be different from the current continuous methods, since it would operate in a different space, that of $\Delta_{\mathcal{Y}}$, which could still be significantly smaller than $\mathbb{R}^d$.

Finally, as noted earlier, representation learning methods solve an inherently harder problem than MIFPO, since such methods need to supply the actual representation, while MIFPO does not. It then would be of interest to understand whether one can use the partial information provided by MIFPO, to guide and improve a full representation learning algorithm.

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

SUPPLEMENTARY MATERIAL

## A   MONOTONICITY OF LOSS UNDER REPRESENTATIONS

As discussed in Section 3, we observe that representations can not increase the performance of the classifier (i.e decrease the loss).

**Lemma A.1.** *For every $(Y, X, A)$, every representation $Z$ as above, and concave $h$,*

$$\mathbb{E}_{z \sim Z} h(\mathbb{P}(Y|Z = z)) \geq \mathbb{E}_{(x,a) \sim (X,A)} h(\mathbb{P}(Y|X = x, A = a)). \tag{23}$$

Note that the right hand-side above can be considered a "trivial" representation, $Z = (X, A)$.

In what follows, to simplify the notation we use expressions of the form $\mathbb{P}(x, a|z)$ to denote the formal expressions $\mathbb{P}(X = x, A = a|Z = z)$, whenever the precise interpretation is clear from context.

*Proof.* For every value $y \in \mathcal{Y}$, we have

$$\mathbb{P}(Y = y|Z = z) = \sum_a \int dx \ \mathbb{P}(Y = y|x, a, z) \frac{\partial \mathbb{P}(x|a, z)}{\partial x} \mathbb{P}(a|z) \tag{24}$$

$$= \sum_a \int dx \ \mathbb{P}(Y = y|x, a) \frac{\partial \mathbb{P}(x|a, z)}{\partial x} \mathbb{P}(a|z) \tag{25}$$

$$= \mathbb{E}_{(x,a) \sim (X,A)|Z=z} \mathbb{P}(Y = y|x, a). \tag{26}$$

Here, on line (24), $\frac{\partial \mathbb{P}(x|a,z)}{\partial x}$ is the density of $\mathbb{P}(x|a, z)$ with respect to $dx$. Crucially, the transition from (24) to (25) is using the property (1). The transition (25) to (26) is a change of notation. Using (26) and the concavity of $h$, we obtain

$$\mathbb{E}_{z \sim Z} h(\mathbb{P}(Y|Z = z)) \geq \mathbb{E}_{z \sim Z} \mathbb{E}_{(x,a) \sim (X,A)|Z=z} h(\mathbb{P}(Y|x, a)) \tag{27}$$

$$= \mathbb{E}_{(x,a) \sim (X,A)} h(\mathbb{P}(Y|X = x, A = a)). \tag{28}$$

$\square$

## B   MIFPO AND OPTIMAL TRANSPORT

In this Section we discuss the relation between the MIFPO minimisation problem, Definition 5.1, and the problem of Optimal Transport (OT). General background on OT may be found in Peyré et al. (2019). We discuss the similarity between OT and the minimisation of (17) under the constraint (19) with $\gamma = 0$, i.e. the perfectly fair case. In this case, (19) is equivalent to the condition $\beta_0(u)r_{u,v,j} = \beta_1(v)r_{v,u,j}$, for all $(u, v, j) \in \mathcal{Z}$. Next, note that thus in is case the expression for $\mathbb{P}(Y|Z = (u, v, j))$ is

$$\frac{\rho_u \alpha_0 \beta_0(u)r_{u,v,j} + \rho_v \alpha_1 \beta_1(v)r_{v,u,j}}{\alpha_0 \beta_0(u)r_{u,v,j} + \alpha_1 \beta_1(v)r_{v,u,j}} = \frac{\rho_u \alpha_0 + \rho_v \alpha_1}{\alpha_0 + \alpha_1}, \tag{29}$$

and *this is independent* of the variables $r$! Therefore we can write the cost (17) as

$$\sum_{u,v} (\alpha_0 + \alpha_1) h\left(\frac{\rho_u \alpha_0 + \rho_v \alpha_1}{\alpha_0 + \alpha_1}\right) \frac{1}{2} \left[\sum_{j \leq k} \beta_0(u)r_{u,v,j} + \beta_1(v)r_{v,u,j}\right]. \tag{30}$$

Note further that for fixed $u, v$, the different $j$'s in this expression play similar roles and could be effectively merged as a single point.

The cost (30) has several similarities with OT. First, in both problems we have two sides, $S_0$ and $S_1$, and we have a certain fixed loss associated with "matching" $u$ and $v$. In case of (30), this loss is $(\alpha_0 + \alpha_1) h\left(\frac{\rho_u \alpha_0 + \rho_v \alpha_1}{\alpha_0 + \alpha_1}\right)$, which describes the information loss incurred by colliding $u$ and $v$ in the representation. And second, similarly to OT, (30) it is *linear* in the variables $r$. Linear programs are conceptually considerably simpler than minimisation of the concave objective (17).

## C  PROOF OF THEOREM 4.1

*Proof.* Assume $T$ is not invertible. Then there is a $z \in \mathcal{Z}$ which has at least two parents in either $S_0$ or $S_1$. Assume without loss of generality that $z$ has two parents in $S_0$. Let

$$U = \{s \in S_0 \mid T_0(z,s) > 0\}, \quad V = \{s \in S_1 \mid T_1(z,s) > 0\} \tag{31}$$

be the sets of parents of $z$ in $S_0$ and $S_1$ respectively. Chose a point $x \in U$, and denote by $U^r = U \setminus \{x\}$ the remainder of $U$. By assumption we have $|U^r| \geq 1$. We also assume that $|V| > 0$. The easier case $|V| = 0$ will be discussed later.

Now, we construct a new representation, $T'$. The range of $T'$ will be $\mathcal{Z}' = \mathcal{Z} \setminus z \cup \{z', z''\}$. That is, we remove $z$ and add two new points. Denote

$$\kappa = \mathbb{P}(x|z, a = 0) = \frac{\beta_0(x)T_0(z,x)}{\sum_{s \in U} \beta_0(s)T_0(z,s)}. \tag{32}$$

Then $T'$ is defined as follows:

$$\begin{cases} T'_a(h,s) = T_a(h,s) & \text{for all } a \in \{0,1\}, \text{ all } s \in S \text{ and all } h \in \mathcal{Z} \setminus \{z\} \\ T'_0(z',x) = T_0(z,x) \\ T'_0(z'',u) = T_0(z,u) & \text{for all } u \in U^r \\ T'_1(z',v) = \kappa T_1(z,v) & \text{for all } v \in V \\ T'_1(z'',v) = (1-\kappa)T_1(z,v) & \text{for all } v \in V. \end{cases} \tag{33}$$

All values of $T'$ that were not explicitly defined in (33) are set to 0. In words, on the side of $S_0$, we move all the parents of $z$ except $x$ to be the parents of $z''$, while $z'$ will have a single parent, $x$. On the $S_1$ side, both $z'$ and $z''$ will have the same parents as $z$, with transitions multiplied by $\kappa$ and $1-\kappa$ respectively. The multiplication by $\kappa$ is crucial for showing both inequalities in (10).

Note that $T'$ can be though of as splitting $z$ into $z'$ and $z''$, such that $z'$ has one parent on the $S_0$ side, and $z''$ has strictly less parents than $z$ had. Once we show that $T'$ satisfies (10), it is clear that by induction we can continue splitting $T'$ until we arrive at an invertible representation which can no longer be split, thus proving the Lemma.

In order to show (10) for $T'$, we will sequentially show the following claims:

$$\mathbb{P}(z) = \mathbb{P}(z') + \mathbb{P}(z'') \text{ and } \mathbb{P}(z') = \kappa \mathbb{P}(z) \tag{34}$$

$$\mathbb{P}(a|z) = \mathbb{P}(a|z') = \mathbb{P}(a|z'') \text{ for } a \in \{0,1\} \tag{35}$$

$$\begin{cases} \text{for } a = 1 & \mathbb{P}(s|z,a) = \mathbb{P}(s|z',a) = \mathbb{P}(s|z'',a) \quad \forall s \in S \\ \text{for } a = 0, s \in U^r & \begin{cases} \mathbb{P}(x|z',a) = 1 & \mathbb{P}(x|z'',a) = 0 \\ \mathbb{P}(s|z',a) = 0 & \mathbb{P}(s|z'',a) = (1-\kappa)^{-1}\mathbb{P}(s|z,a) \end{cases} \end{cases} \tag{36}$$

$$\mathbb{P}(Y|z) = \kappa \mathbb{P}(Y|z') + (1-\kappa)\mathbb{P}(Y|z'') \tag{37}$$

$$\mathbb{P}(z) h(\mathbb{P}(Y|z)) \geq \mathbb{P}(z') h(\mathbb{P}(Y|z')) + \mathbb{P}(z'') h(\mathbb{P}(Y|z'')) \tag{38}$$

$$|\mu_0(z) - \mu_1(z)| = |\mu'_0(z') - \mu'_1(z')| + |\mu'_0(z'') - \mu'_1(z'')|. \tag{39}$$

Here the probabilities involving $z', z''$ refer to the representation $T'$. Observe that the left hand side of (38) is the contribution of $z$ to the performance cost $\mathbb{E}_{t \sim Z} h(\mathbb{P}(Y|Z=t))$ of $T$, while the right hand side of (38) is the contribution of $z', z''$ to the performance cost of $T'$. Since all other elements $t \in \mathcal{Z}$ have identical contributions, this shows the first inequality in (10). Similarly, recall that

$$\|\mu_0 - \mu_1\|_{TV} = \frac{1}{2} \sum_{t \in \mathcal{Z}} |\mu_0(t) - \mu_1(t)|, \tag{40}$$

and thus the left hand side of (39) is the contribution of $z$ to $\|\mu_0 - \mu_1\|_{TV}$, with the right hand side being the contribution of $z', z''$ to $\|\mu'_0 - \mu'_1\|_{TV}$, therefore yielding the claim $\|\mu'_0 - \mu'_1\|_{TV} = \|\mu_0 - \mu_1\|_{TV}$.

**Claim (34):** By definition,

$$\mathbb{P}\left(z'\right) = \alpha_0 \beta_0(x) T_0'(z', x) + \alpha_1 \sum_{s \in V} \beta_1(s) T_1'(z', s) \tag{41}$$

$$= \alpha_0 \beta_0(x) T_0(z, x) + \kappa \alpha_1 \sum_{s \in V} \beta_1(s) T_1(z, s) \tag{42}$$

$$= \kappa \alpha_0 \sum_{s \in U} \beta_0(s) T_0(z, s) + \kappa \alpha_1 \sum_{s \in V} \beta_1(s) T_1(z, s) \tag{43}$$

$$= \kappa \mathbb{P}\left(z\right). \tag{44}$$

Similarly, by definition we have

$$\mathbb{P}\left(z''\right) = \alpha_0 \sum_{s \in U^r} \beta_0(s) T_0(z, s) + (1 - \kappa) \kappa \alpha_1 \sum_{s \in V} \beta_1(s) T_1(z, s), \tag{45}$$

and summing this with (42), we obtain $\mathbb{P}\left(z\right) = \mathbb{P}\left(z'\right) + \mathbb{P}\left(z''\right)$.

**Claim (35):** Note that it is sufficient to prove the claim for $a = 0$ since the probabilities sum to 1. Write

$$\mathbb{P}\left(a = 0 | z\right) = \frac{\mathbb{P}\left(a = 0, z\right)}{\mathbb{P}\left(z\right)} \tag{46}$$

$$= \frac{\alpha_0 \sum_{s \in U} \beta_0(s) T_0(z, s)}{\mathbb{P}\left(z\right)} \tag{47}$$

$$= \frac{\kappa \cdot \alpha_0 \sum_{s \in U} \beta_0(s) T_0(z, s)}{\kappa \cdot \mathbb{P}\left(z\right)} \tag{48}$$

$$= \frac{\alpha_0 \beta_0(x) T_0(z, x)}{\mathbb{P}\left(z'\right)} \tag{49}$$

$$= \mathbb{P}\left(a = 0 | z'\right). \tag{50}$$

Similarly,

$$\mathbb{P}\left(a = 0 | z\right) = \frac{\mathbb{P}\left(a = 0, z\right)}{\mathbb{P}\left(z\right)} \tag{51}$$

$$= \frac{\alpha_0 \sum_{s \in U} \beta_0(s) T_0(z, s)}{\mathbb{P}\left(z\right)} \tag{52}$$

$$= \frac{(1 - \kappa) \cdot \alpha_0 \sum_{s \in U} \beta_0(s) T_0(z, s)}{(1 - \kappa) \cdot \mathbb{P}\left(z\right)} \tag{53}$$

$$= \frac{\alpha_0 \sum_{s \in U^r} \beta_0(s) T_0(z, s)}{\mathbb{P}\left(z''\right)} \tag{54}$$

$$= \mathbb{P}\left(a = 0 | z''\right). \tag{55}$$

**Claim (36):** For $a = 1$, let us show $\mathbb{P}\left(s | z, a\right) = \mathbb{P}\left(s | z', a\right)$.

$$\mathbb{P}\left(s | z, a = 1\right) = \frac{\alpha_1 \beta_1(s) T_1(z, s)}{\alpha_1 \sum_{s' \in V} \beta_1(s') T_1(z, s')} \tag{56}$$

$$= \frac{\kappa \alpha_1 \beta_1(s) T_1(z, s)}{\kappa \alpha_1 \sum_{s' \in V} \beta_1(s') T_1(z, s')} \tag{57}$$

$$= \frac{\alpha_1 \beta_1(s) T_1'(z, s)}{\alpha_1 \sum_{s' \in V} \beta_1(s') T_1'(z, s')} \tag{58}$$

$$= \mathbb{P}\left(s | z', a = 1\right). \tag{59}$$

The statement $\mathbb{P}\left(s | z, a\right) = \mathbb{P}\left(s | z'', a\right)$ is shown similarly. Next, for $a = 0$, we have $\mathbb{P}\left(x | z', a\right) = 1$ and $\mathbb{P}\left(x | z'', a\right) = 0$ by the definition of the coupling $T'$. Moreover, for $s \in U^r$, $\mathbb{P}\left(s | z', a\right) = 0$ also

follows by the definition of $T'$. Finally, write

$$\mathbb{P}\left(s|z'', a=0\right) = \frac{\alpha_0 \beta_0(s) T_0'(z'', s)}{\sum_{s \in U^r} \alpha_0 \beta_0(s') T_0'(z'', s')} \tag{60}$$

$$= \frac{\alpha_0 \beta_0(s) T_0(z, s)}{\sum_{s \in U^r} \alpha_0 \beta_0(s') T_0(z, s')} \tag{61}$$

$$= \frac{\alpha_0 \beta_0(s) T_0(z, s)}{(1 - \kappa) \sum_{s \in U} \alpha_0 \beta_0(s') T_0(z, s')} \tag{62}$$

$$= (1 - \kappa)^{-1} \mathbb{P}\left(s|z, a=0\right). \tag{63}$$

**Claim** (37)**:** We first observe that for any representation (and any z),

$$\mathbb{P}\left(Y = y | Z = z\right) = \frac{\sum_{s,a} \mathbb{P}\left(Y = y, s, a, z\right)}{\mathbb{P}\left(z\right)} \tag{64}$$

$$= \frac{\sum_{s,a} \mathbb{P}\left(Y = y | s, a\right) \mathbb{P}\left(s, a, z\right)}{\mathbb{P}\left(z\right)} \tag{65}$$

$$= \sum_a \mathbb{P}\left(a|z\right) \left[\sum_{s \in S_a} \mathbb{P}\left(Y = y | s, a\right) \mathbb{P}\left(s|a, z\right)\right], \tag{66}$$

where we have used the property (1) for the transition (64)-(65). Now, using (36), for $a = 1$ we have

$$\sum_{s \in S_1} \mathbb{P}\left(Y = y | s, a = 1\right) \mathbb{P}\left(s|a = 1, z\right) = \sum_{s \in S_1} \mathbb{P}\left(Y = y | s, a = 1\right) \mathbb{P}\left(s|a = 1, z'\right) = \sum_{s \in S_1} \mathbb{P}\left(Y = y | s, a = 1\right) \mathbb{P}\left(s|a = 1, z''\right). \tag{67}$$

For $a = 0$, we have for $z'$ using (36):

$$\sum_{s \in S_0} \mathbb{P}\left(Y = y | s, a = 1\right) \mathbb{P}\left(s|a = 1, z'\right) = \mathbb{P}\left(Y = y | x, a = 1\right). \tag{68}$$

For $a = 0$ and $z''$ we have

$$\sum_{s \in S_0} \mathbb{P}\left(Y = y | s, a = 1\right) \mathbb{P}\left(s|a = 1, z''\right) = \sum_{s \in U^r} \mathbb{P}\left(Y = y | s, a = 1\right) \mathbb{P}\left(s|a = 1, z''\right) \tag{69}$$

$$= (1 - \kappa)^{-1} \sum_{s \in U^r} \mathbb{P}\left(Y = y | s, a = 1\right) \mathbb{P}\left(s|a = 1, z\right) \tag{70}$$

where we have used (36) again on the last line.

Combining (67),(68),(70), and using (35) and the general expression (66), we obtain the claim (37).

**Claim** (38)**:** This follows immediately from (37) by using (34) and the concavity of $h$.

**Claim** (39)**:** By definition, for every representation, $\mu_a(z) = \mathbb{P}\left(Z = z|a\right) = \frac{\mathbb{P}(z|a)\mathbb{P}(z)}{\mathbb{P}(a)}$. Thus, using (34),(35) we have for $a \in \{0, 1\}$,

$$\mu_a(z') = \kappa \mu_a(z) \text{ and } \mu_a(z'') = (1 - \kappa)\mu_a(z), \tag{71}$$

which in turn yields (39).

It remains only to recall that we have derived (38),(39) under the assumption that $|V| > 0$. That is, we assumed that the point $z$ which fails invertability on $S_0$ has some parents in $S_1$. The case when $|V| = 0$, i.e. there are no parents in $S_1$ can be treated using a similar argument, but is much simpler. Indeed, in this case one can simply split $z$ into $z'$ and $z''$ and splitting the $S_0$ weight between them as before, without the need to carefully balance the interaction of probabilities with $S_1$ via $\kappa$.

$\square$

## D UNIFORM APPROXIMATION AND TWO POINT REPRESENTATIONS

As discussed in Sections 1,5.2, we are interested in showing that all optimal invertible representations, no matter which, and no matter on which set $\mathcal{Z}'$, can be approximated using a representation with the following property: For every $u \in S_0, v \in S_1$, there are at most $k$ points $z \in \mathcal{Z}$ that have $(u, v)$ as parents, see Figure 2(a). Here $k$ would depend only on the desired approximation degree, but not on $\mathcal{Z}'$, or on the exact representation we are approximating. We therefore refer to this result as the Uniform Approximation result. Its implications for practical use were discussed in Section 5.2.

To proceed with the analysis, in what follows we introduce the notion of two-point representation. The main result is given as Lemma D.1 below.

This Section uses the notation of Section 4. Let $T$ be an invertible representation, let $u \in S_0, v \in S_1$ be some points, and denote by $\mathcal{Z}_{uv} = \left\{ z^j \right\}_1^k$ the set of all points $z \in \mathcal{Z}$ which have $u$ and $v$ as parents. Denote by

$$w_u = \sum_{j=1}^k \beta_0(u) T_0(z^j, u) \text{ and } w_v = \sum_{j=1}^k \beta_1(v) T_1(z^j, v) \tag{72}$$

the total weights of $\beta_0$ and $\beta_1$ transferred by the representation from $u$ and $v$ respectively to $\mathcal{Z}_{uv}$. Recall that $\rho_u, \rho_v$ denote the distributions of $Y$ conditioned on $u, v$. We call the situation above, i.e. the collection of numbers $\left( \left\{ \beta_0(u) T_0(z^j, u) \right\}_{j \leq k}, \left\{ \beta_1(v) T_1(Z^j, v) \right\}_{j \leq k} \right)$, a *two point representation*, since it describes how the weight from the points $u, v$ is distributed in the representation, independently of the rest of the representation. The contribution of $\mathcal{Z}_{uv}$ to the global performance cost is

$$E_{uv,T} := \sum_{j \leq k} \mathbb{P}\left(z^j\right) h(\mathbb{P}\left(Y | z^j\right)) \tag{73}$$

$$= \sum_{j \leq k} \left( \alpha_0 \beta_0(u) T_0(z^j, u) + \alpha_1 \beta_1(v) T_1(z^j, v) \right) h\left( \frac{\alpha_0 \beta_0(u) T_0(z^j, u) \rho_u + \alpha_1 \beta_1(v) T_1(z^j, v) \rho_v}{\alpha_0 \beta_0(u) T_0(z^j, u) + \alpha_1 \beta_1(v) T_1(z^j, v)} \right), \tag{74}$$

while its contribution to the fairness condition is

$$F_{uv,T} = \frac{1}{2} \sum_{j \leq k} \left| \beta_0(u) T_1(z^j, u) - \beta_1(v) T_1(z^j, v) \right|. \tag{75}$$

Let us now consider two extreme cases of two-point representations. Assume that the total amounts of weight to be represented, $w_u, w_v$ are fixed. The first case is when $k = 1$, and this is the maximum fairness case, since in this case the weights $w_u, w_v$ overlap as much as possible. Indeed, the contributions to the fairness penalty and performance cost in this case are

$$|w_u - w_v| \text{ and } (\alpha_0 w_u + \alpha_1 w_v) h\left( \frac{\alpha_0 w_u \rho_u + \alpha_1 w_v \rho_v}{\alpha_0 w_u + \alpha_1 w_v} \right) \tag{76}$$

respectively. The other extreme case is when $w_u$ and $w_v$ do not overlap at all. This case be realised with $k = 2$, by sending all $w_u$ to $z^1$ and all $w_v$ to $z^2$. The fairness and performance contributions would be

$$w_u + w_v \text{ and } \alpha_0 w_u \cdot h\left(\rho_u\right) + \alpha_1 w_v \cdot h\left(\rho_v\right), \tag{77}$$

respectively. Note that the fairness penalty is the maximum possible, while the performance cost is the minimum possible (indeed, this is the cost before the representation, and any representation can only increase it, by Lemma A.1). We thus observed that each two points $u, v$, with fixed total weight $w_u, w_v$, can have their own Pareto front of performance-fairness. One could, in principle, fix a threshold $\gamma_{uv}, |w_u - w_v| \leq \gamma_{uv} \leq w_u + w_v$ for the fairness penalty (75), and obtain a performance cost between that in (76) and (77). However, it is not clear how large the number of points $k$ should be in order to realise such intermediate representations. In the following Lemma we show that one can uniformly approximate all the points on the two-point Pareto front using a fixed number of points, that depends only on the function $h$. This means that in practice one can choose a certain number $n$ of $z$ points, and have guaranteed bounds on the possible amount of loss incurred with respect to all representations of all other sizes.

**Lemma D.1.** *For every $\varepsilon > 0$, there a number $n = n_\varepsilon$ depending only on the function $h$, with the following property: For every two-point representation $\left(\{\beta_0(u)T_0(z^j,u)\}_{j\le k}, \{\beta_1(v)T_1(Z^j,v)\}_{j\le k}\right)$, with total weights $w_u, w_v$, there is a two point representation $T'$ on a set $\mathcal{Z}'_{u,v}$, with the same total weights, such that $|\mathcal{Z}'_{uv}| \le n$, and such that*

$$F_{uv,T'} \le F_{uv,T} \text{ and } E_{uv,T'} \le E_{uv,T} + 2(w_u + w_v)\varepsilon. \tag{78}$$

*Proof.* To aid with brevity of notation, define for $j \le k$

$$c_0^j = \alpha_0\beta_0(u)T_0(z^j,u), \quad c_1^j = \alpha_1\beta_1(v)T_1(z^j,v). \tag{79}$$

Then we can write

$$E_{uv,T} = \sum_{j\le k}(c_0^j + c_1^j) \cdot h\left(\frac{c_0^j\rho_u + c_1^j\rho_v}{c_0^j + c_1^j}\right), \quad F_{uv,T} = \frac{1}{2}\sum_{j\le k}\left|\Lambda c^j\right|, \tag{80}$$

where $\Lambda$ is the vector $\Lambda = (\alpha_0^{-1}, -\alpha_1^{-1})$, $c^j = (c_0^j, c_1^j)$, and $\Lambda c^j$ is the inner product of the two.

Observe that the cost $E_{uv,T}$ depends on $c^j$ mainly through the fractions $\frac{c_0^j}{c_0^j + c_1^j}$. Our strategy thus would be to approximate all $k$ of such fractions by a $\delta$-net of a size independent of $k$. To this end, set

$$p^j = \frac{c_0^j}{c_0^j + c_1^j} \tag{81}$$

and define $h_{uv}: [0,1] \to \mathbb{R}$ by

$$h_{uv}(p) = h(p\rho_u + (1-p)\rho_v). \tag{82}$$

Since $h$ is continuous (by assumption), and defined on a compact set, it is *uniformly* continuous, and so is $h_{uv}$. By definition, this means there is a $\delta > 0$ such that for all $p, p'$ with $|p - p'| \le \delta$, it holds that $|h_{uv}(p) - h_{uv}(p')| \le \varepsilon$. Let us now choose $\{x_i\}_{i=1}^n$ to be a $\delta$ net on $[0,1]$. For every $i \le n$ set

$$\Gamma_i = \left\{j \mid \left|p^j - x_i\right| \le \delta, \text{ and } i \text{ is minimal with this property}\right\}. \tag{83}$$

That is, $\Gamma_i$ is the set of indices $j$ such that $p^j$ is approximated by $x^i$. Using $x_i$ we construct the representation $T'$ as follows: For $a \in \{0, 1\}$ set

$$c_a^{\prime i} = \sum_{j \in \Gamma_i} c_a^j. \tag{84}$$

For $n$ new points, $z^i \in \mathcal{Z}'_{uv}$, set $T_0'(z'^i, u) = c_0'^i/\beta_0(u)$, $T_1'(z'^i, v) = c_1'^i/\beta_1(v)$. Note that the total weights are preserved, $\sum_{i\le n}c_0'^i = w_u$ and $\sum_{i\le n}c_1'^i = w_v$.

Next, for every $j \in \Gamma_i$ we have

$$\left|\frac{c_0^j}{c_0^j + c_1^j} - x_i\right| \le \delta. \tag{85}$$

Thus

$$\left|\frac{c_0'^i}{c_0'^i + c_1'^i} - x_i\right| = \left|\frac{\sum_{j\in\Gamma_j}\left[c_0^j - (c_0^j + c_1^j)x_i\right]}{c_0'^i + c_1'^i}\right| \le \frac{\sum_{j\in\Gamma_j}\delta(c_0^j + c_1^j)}{c_0'^i + c_1'^i} = \delta. \tag{86}$$

Next, observe that by the construction of $x_i$,

$$\left|E_{uv,T} - \sum_i(c_0'^i + c_1'^i)h_{uv}(x_i)\right| = \left|\sum_i\sum_{j\in\Gamma_i}(c_0^j + c_1^j)h_{uv}(p^j) - \sum_i(c_0'^i + c_1'^i)h_{uv}(x_i)\right| \tag{87}$$

$$\le \sum_i\sum_{j\in\Gamma_i}(c_0^j + c_1^j)\varepsilon \tag{88}$$

$$= (w_u + w_v)\varepsilon. \tag{89}$$

In addition,

$$\left| E_{uv,T'} - \sum_i (c_0'^i + c_1'^i) h_{uv}(x_i) \right| = \left| \sum_i (c_0'^i + c_1'^i) h_{uv}\left(\frac{c_0'^i}{c_0'^i + c_1'^i}\right) - \sum_i (c_0'^i + c_1'^i) h_{uv}(x_i) \right| \quad (90)$$

$$\leq (w_u + w_v)\varepsilon, \quad (91)$$

where we have used (86) in the last transition.

Combining the two inequalities yields the second part of (78),

$$|E_{uv,T'} - E_{uv,T}| \leq 2(w_u + w_v)\varepsilon. \quad (92)$$

Finally, note that

$$\sum_i \left| \Lambda c'^i \right| = \sum_i \left| \sum_{j \in \Gamma_j} \Lambda c^j \right| \quad (93)$$

$$\leq \sum_i \sum_{j \in \Gamma_j} \left| \Lambda c^j \right| \quad (94)$$

$$= \sum_j \left| \Lambda c^j \right|, \quad (95)$$

yielding the first part of, and thus completing the proof of, statement (78).

It remains to observe that above we have used a $\delta$ net for $h_{uv}$, which depends on $\rho_u, \rho_v$. However, we can directly build an appropriate $\delta$-net in full range of $h$, the simplex $\Delta_{\mathcal{Y}}$, which would produce bounds valid for all $u, v$. Indeed, let $\delta'$ be such that $|h(\nu) - h(\nu)| \leq \varepsilon$ for all $\mu, \nu \in \Delta_{\mathcal{Y}}$ with $\|u - v\|_1 \leq \delta'$. Observe that the map $p \mapsto p\rho_v + (1 - p)\rho_u$ is 2-Lipschitz from $\mathbb{R}$ to $\Delta_{\mathcal{Y}}$ equipped with the $\|\cdot\|_1$ norm, for any $u, v \in \Delta_{\mathcal{Y}}$. Thus, choosing $\delta = \frac{1}{2}\delta'$, we have $|h_{uv}(p) - h_{uv}(p')| \leq \varepsilon$ if $|p - p'| \leq \delta$. This completes the proof of the Lemma. $\qquad\square$

## E   CONCAVITY OF $E_r$

Note that the variables $r$ appear in (12) both as coefficients multiplying $h$ and inside the arguments of $h$, in a fairly involved manner. Nevertheless, the cost turns out to still retain an interesting structure, as it is *concave*, if $h$ is. We record this in the following Lemma.

**Lemma E.1.** *If $h : \Delta_{\mathcal{Y}} \to \mathbb{R}$ is concave, then of every $\rho_1, \rho_2 \in \Delta_{\mathcal{Y}}$ the function $g : \mathbb{R}^2 \to \mathbb{R}$, given by $g((c_1, c_2)) = (c_1 + c_2)h(\frac{c_1\rho_1 + c_2\rho_2}{c_1 + c_2})$ is concave.*

*Proof.* It is sufficient to show that for every $c, c' \in \mathbb{R}^2$, we have $g((c + c')/2) \geq \frac{1}{2}(g(c) + g(c'))$. To this end, define the map $F : \mathbb{R}^2 \to \Delta_{\mathcal{Y}}$ by

$$F(c) = \frac{c_1\rho_1 + c_2\rho_2}{c_1 + c_2} \quad (96)$$

and note that

$$F((c + c')/2) = F(c)\frac{c_1 + c_2}{c_1 + c_2 + c_1' + c_2'} + F(c')\frac{c_1' + c_2'}{c_1 + c_2 + c_1' + c_2'}. \quad (97)$$

It then follows that

$$g((c + c')/2) = \frac{1}{2}(c_1 + c_2 + c_1' + c_2')h(F((c + c')/2)) \quad (98)$$

$$= \frac{1}{2}(c_1 + c_2 + c_1' + c_2')h\left( F(c)\frac{c_1 + c_2}{c_1 + c_2 + c_1' + c_2'} + F(c')\frac{c_1' + c_2'}{c_1 + c_2 + c_1' + c_2'} \right) \quad (99)$$

$$\geq \frac{1}{2}(c_1 + c_2 + c_1' + c_2')\left[ \frac{c_1 + c_2}{c_1 + c_2 + c_1' + c_2'}h(F(c)) + \frac{c_1' + c_2'}{c_1 + c_2 + c_1' + c_2'}h(F(c')) \right] \quad (100)$$

$$= \frac{1}{2}(g(c) + g(c')). \quad (101)$$

$$\square$$

# F  ADDITIONAL PROOFS

## F.1  PROOF OF LEMMA 5.1

*Proof.* For this proof it is more convenient to work with the $\ell_1$ norm $\|\cdot\|_1$ directly. Recall that $\|\mu_0 - \mu_1\|_{TV} = \frac{1}{2}\|\mu_0 - \mu_1\|_1$ and that

$$\|\mu_0 - \mu_1\|_1 = \sum_z |\mu_0(z) - \mu_1(z)|. \tag{102}$$

Assume that $\|\mu_0 - \mu_1\|_1 = 2\gamma$. Define the functions $\bar{\phi}_0(z) = \mathbb{1}_{\{\mu_1 \geq \mu_0\}}(z) \cdot (\mu_1(z) - \mu_0(z))$ and $\bar{\phi}_1(z) = \mathbb{1}_{\{\mu_0 \geq \mu_1\}}(z) \cdot (\mu_0(z) - \mu_1(z))$. Note that we then have

$$\sum_z \bar{\phi}_0(z) = \sum_z \bar{\phi}_1(z) = \gamma. \tag{103}$$

Indeed, define

$$\eta(z) = \mathbb{1}_{\{\mu_0 \geq \mu_1\}}(z) \cdot \mu_1(z) + \mathbb{1}_{\{\mu_1 \geq \mu_0\}}(z) \cdot \mu_0(z). \tag{104}$$

Clearly, $\eta + \bar{\phi}_0 = \mu_1$ and thus $\sum_z \eta(z) + \bar{\phi}_0(z) = 1$. Similarly, $\sum_z \eta(z) + \bar{\phi}_1(z) = 1$. Therefore we have

$$\sum_z \bar{\phi}_0(z) = \sum_z \bar{\phi}_1(z). \tag{105}$$

Note also that we can write

$$2\gamma = \|\mu_0 - \mu_1\|_1 = \sum_z |\mu_0(z) - \mu_1(z)| = \sum_z \left(\bar{\phi}_0(z) + \bar{\phi}_1(z)\right), \tag{106}$$

which combined with (105) yields (103).

Next, we can also directly verify that

$$\mu_0 + \bar{\phi}_0 = \mu_1 + \bar{\phi}_1, \tag{107}$$

and thus setting $\phi_0 = \gamma^{-1}\bar{\phi}_0, \phi_1 = \gamma^{-1}\bar{\phi}_1$ completes the proof of this direction.

In the other direction, given $\phi_0, \phi_1 \in \Delta_{\mathcal{Z}}$ such that $\mu_0 + \gamma\phi_0 = \mu_1 + \gamma\phi_1$, we have

$$\sum_z |\mu_0(z) - \mu_1(z)| = \gamma \sum_z |\phi_1(z) - \phi_0(z)| \leq 2\gamma, \tag{108}$$

thus completing the proof. $\qquad\square$

## F.2  THE FACTOR REPRESENTATION

Fix $z \in \mathcal{Z}$. To show $\mu_0(z) = \bar{\mu}_0(z)$ write, by definition,

$$\bar{\mu}_0(z) = \sum_\rho \bar{\beta}_0(\rho)\bar{T}_0(z, \rho) \tag{109}$$

$$= \sum_\rho \sum_{v \in \Psi^{-1}(\{\rho\})} \beta_0(v)T_0(z, v) \tag{110}$$

$$= \sum_{v \in S_0} \beta_0(v)T_0(z, v) \tag{111}$$

$$= \mu_0(z). \tag{112}$$

The case $a = 1$ is shown similarly. Next, the above also implies that

$$\mathbb{P}_{\bar{T}}(z) = \alpha_0\bar{\mu}_0(z) + \alpha_1\bar{\mu}_1(z) = \alpha_0\mu_0(z) + \alpha_1\mu_1(z) = \mathbb{P}_T(z). \tag{113}$$

Finally, by definition we have

$$\mathbb{P}_{\bar{T}}(Y|z) = \sum_{a \in \{0,1\}} \sum_\rho \mathbb{P}_{\bar{T}}(Y|\rho, a)\,\mathbb{P}_{\bar{T}}(\rho, a|z) \tag{114}$$

$$= \mathbb{P}_{\bar{T}}(z)^{-1} \sum_{a \in \{0,1\}} \sum_\rho \mathbb{P}_{\bar{T}}(Y|\rho, a)\,\mathbb{P}_{\bar{T}}(z|\rho, a)\,\bar{\beta}_a(\rho)\alpha_a \tag{115}$$

Taking (113) into account, to show $\mathbb{P}_{\bar{T}}(Y|z) = \mathbb{P}_T(Y|z)$ it thus suffices to show that for a fixed $a$,

$$\sum_\rho \mathbb{P}_{\bar{T}}(Y|\rho, a)\,\mathbb{P}_{\bar{T}}(z|\rho, a)\,\bar{\beta}_a(\rho)\alpha_a = \sum_x \mathbb{P}_T(Y|x, a)\,\mathbb{P}_T(z|x, a)\,\beta_a(\rho)\alpha_a. \quad (116)$$

Let us consider $a = 0$. Then

$$\sum_\rho \mathbb{P}_{\bar{T}}(Y|\rho, a=0)\,\mathbb{P}_{\bar{T}}(z|\rho, a=0)\,\bar{\beta}_0(\rho)\alpha_0 = \alpha_0 \sum_\rho \rho\bar{T}_0(z, \rho)\bar{\beta}_0(\rho) \quad (117)$$

$$= \alpha_0 \sum_\rho \rho \sum_{v \in \Psi^{-1}(\{\rho\})} \beta_0(v)T_0(z, v) \quad (118)$$

$$= \alpha_0 \sum_{v \in S_0} \rho_v \beta_0(v)T_0(z, v). \quad (119)$$

where we have use the definitions on line (117), and in particular that $\mathbb{P}_{\bar{T}}(Y|\rho, a) = \rho$ by construction. This shows (116) for $a = 0$, and the case $a = 1$ is shown similarly, thus completing the proof.

## G   EXPERIMENTS - ADDITIONAL DETAILS

### G.1   DATASETS FOR EVALUATIONS

The FNF data preprocessing, as implemented in Balunović et al. (2022b), was used for both MIFPO and FNF models. In running FNF, we have used the original code and configuration from Balunović et al. (2022b), with the exception that we have disabled the noise injection into features, and the subsequent logit transformation. This was disabled since this step was not a part of preprocessing, and due its particular implementation in Balunović et al. (2022b), it significantly complicated the comparison with an external model like XGBoost. This step was also performed during both training and *evaluation*, thus producing a less clean experiment, and was not documented anywhere in the paper Balunović et al. (2022a). We note that retaining this step would not have changed the FNF related analysis or its conclusions, as presented in Section 6.

### G.2   MINIMIZATION OF CONCAVE FUNCTIONS UNDER CONVEX CONSTRAINTS

As described in figure 6, we used the disciplined convex concave programming (DCCP) framework and the associated solver, (Shen et al., 2016; 2024) for solving the concave minimization with convex constraints problem.

Minimizing concave functions under convex constraints is a common problem in optimization theory. Unlike convex optimization where global minima can be readily found, in concave minimization problems we only know that the local minimas lie on the boundaries of the feasible region defined by the convex constraints. While techniques such as branch-and-bound algorithms, cutting plane methods, and heuristic approaches are often employed, here we used the framework of DCCP which gain a lot of popularity in recent years.

The DCCP framework extends disciplined convex programming (DCP) to handle nonconvex problems with objective and constraint functions composed of convex and concave terms. The idea behind a "disciplined" methodology for convex optimization is to impose a set of conventions inspired by basic principles of convex analysis and the practices of convex optimization experts. These "disciplined" conventions, while simple and teachable, allow much of the manipulation and transformation required for analyzing and solving convex programs to be automated. DCCP builds upon this idea, providing an organized heuristic approach for solving a broader class of nonconvex problems by combining DCP principles with convex-concave programming (CCP) methods, and is implemented as an extension to the CVXPY package in Python.

While convenient, the use of the disciplined framework bears some limitations. Mainly, generic operations like element-wise multiplication are not under the allowed set of operations (and for obvious reasons), which limits the usability. Notice, that for the prediction accuracy measure $h(p) = \min(p, 1-p)$ this is not a problem, but for the entropy classification error $h((1-p, p)) = -p\log p - (1-p)\log(1-p)$, this is more challenging. Nevertheless, here we show that the standard DCCP framework allows for entropy classification error.

**Lemma G.1.** *Let $a = (1 - \alpha)\mathcal{V}(v)r_{v,z}$ and $b = \alpha \cdot \mathcal{U}(u)r_{u,z}$, with $p_v = p_a$ and $p_u = p_b$.*

*We can write our cost function under the entropy accuracy error as:*

$$(a + b) \cdot \text{entropy}\left(\frac{a \cdot p_a + b \cdot p_b}{a + b}\right)$$
$$= \text{entropy}(a \cdot (1 - p_a) + b \cdot (1 - p_b)) + \text{entropy}(a \cdot p_a + b \cdot p_b) - \text{entropy}(a + b)$$

*Proof.*

$$\text{entropy}(x) = \text{entropy}(x, 1 - x) = x\left[\log(x) - \log(1 - x)\right] + \log(1 - x)$$

Thus,

$$\text{entropy}\left(\frac{a \cdot p_a + b \cdot p_b}{a + b}\right) = \frac{a \cdot p_a + b \cdot p_b}{a + b}\left[\log\left(\frac{a \cdot p_a + b \cdot p_b}{a + b}\right) - \log\left(1 - \frac{a \cdot p_a + b \cdot p_b}{a + b}\right)\right]$$
$$+ \log\left(1 - \frac{a \cdot p_a + b \cdot p_b}{a + b}\right)$$
$$= \frac{a \cdot p_a + b \cdot p_b}{a + b}\left[\log(a \cdot p_a + b \cdot p_b) - \log(a + b - a \cdot p_a - b \cdot p_b)\right]$$
$$+ \log(a + b - a \cdot p_a - b \cdot p_b) - \log(a + b)$$

Hence :

$$(a + b) \cdot \text{entropy}\left(\frac{a \cdot p_a + b \cdot p_b}{a + b}\right) = (a + b) \cdot \left[\frac{a \cdot p_a + b \cdot p_b}{a + b}\left[\log(a \cdot p_a + b \cdot p_b) - \log(a + b - a \cdot p_a - b \cdot p_b)\right]\right]$$
$$+ (a + b) \cdot \log(a + b - a \cdot p_a - b \cdot p_b)$$
$$- (a + b) \cdot \log(a + b)$$

$$= \text{entropy}(a \cdot p_a + b \cdot p_b) - (a \cdot p_a + b \cdot p_b) \cdot \log(a + b - a \cdot p_a - b \cdot p_b)$$
$$+ (a + b) \cdot \log(a + b - a \cdot p_a - b \cdot p_b) - \text{entropy}(a + b)$$

Finally, the expression can be written as:

$$= \text{entropy}(a + b - a \cdot p_a - b \cdot p_b) + \text{entropy}(a \cdot p_a + b \cdot p_b) - \text{entropy}(a + b)$$

$$= \text{entropy}(a \cdot (1 - p_a) + b \cdot (1 - p_b)) + \text{entropy}(a \cdot p_a + b \cdot p_b) - \text{entropy}(a + b)$$

$\square$

Given the element-wise entropy function $x \cdot (1 - x)$ is with known characteristics and under the dccp framework, we can use the entropy error for our cost using :

$$= \text{entropy}((1 - \alpha)\mathcal{V}(v)r_{v,z} \cdot (1 - p_v) + \alpha \cdot \mathcal{U}(u)r_{u,z} \cdot (1 - p_u))$$
$$- \text{entropy}((1 - \alpha)\mathcal{V}(v)r_{v,z} \cdot p_v + \alpha \cdot \mathcal{U}(u)r_{u,z} \cdot p_u)$$
$$- \text{entropy}((1 - \alpha)\mathcal{V}(v)r_{v,z} + \alpha \cdot \mathcal{U}(u)r_{u,z})$$

### G.3 IMPLEMENTATIONS AND COMPUTATIONAL DETAILS

The Pareto front evaluation requires two main parts - solving the optimization problem described above, and building a calibrated classifier required for evaluating $c_a = P(Y|X, A = a)$ (see Algorithm 1).

For a calibrated classifier, we are using standard model calibration. Model calibration is a well-studied problem where we fit a monotonic function to the probabilities of some base model so that the probabilities will reflect real probabilities, that is, $P(Y|X)$. Here, we used Isotonic regression

(Berta et al., 2024) for model calibration with XGBoost (Chen et al., 2015) as the base model. For training the XGBoost model, a GridSearchCV approach is employed to find the best hyperparameters from a specified parameter grid, using 3-fold cross-validation.

The experiments were conducted on a system with an Intel Core i9-12900KS CPU (16 cores, 24 threads), 64 GB of RAM, and an NVIDIA GeForce RTX 3090 GPU.

## H  FAIR CLASSIFIERS AS FAIR REPRESENTATIONS

As discussed in Section 3, Pareto front of binary classifiers with statistical parity can be computed from the Pareto front of representations with total variation fairness distance. In this Section we provide the proof of this result, Lemma 3.1. The Lemma and the related notation are restated here for convenience.

For a binary classifier $\hat{Y}$ of $Y$, its prediction error is defined as usual by $\epsilon(\hat{Y}) := \mathbb{P}\left(\hat{Y} \neq Y\right)$. The statistical parity *distance* of $\hat{Y}$ is defined as

$$\Delta_{SP}(\hat{Y}) := \left|\mathbb{P}\left(\hat{Y} = 1 | A = 1\right) - \mathbb{P}\left(\hat{Y} = 1 | A = 0\right)\right|. \tag{120}$$

Let the uncertainy measure $h$ be defined by (4). Note that the first part of the Lemma does use the special properties of this $h$ and does not necessarily hold for other costs $h$.

**Lemma H.1.** *Let $\hat{Y}$ be a classifier of $Y$. Then there is a representation given by a random variable $Z$ on a set $\mathcal{Z}$ with $|\mathcal{Z}| = 2$, such that*

$$\mathbb{E}_{z \sim Z} h(\mathbb{P}\left(Y | Z = z\right)) \leq \epsilon(\hat{Y}) \text{ and } \|\mu_0 - \mu_1\|_{TV} \leq \Delta_{SP}(\hat{Y}). \tag{121}$$

*Conversely, for any given representation $Z$, there is a classifier $\hat{Y}$ of $Y$ as a function of $Z$ (and thus of $(X, A)$), such that*

$$\epsilon(\hat{Y}) \leq \mathbb{E}_{z \sim Z} h(\mathbb{P}\left(Y | Z = z\right)) \text{ and } \Delta_{SP}(\hat{Y}) \leq \|\mu_0 - \mu_1\|_{TV}. \tag{122}$$

*Proof.* Let us begin with the second part of the Lemma, inequalities (122). Given a representation $Z$, $\epsilon(\hat{Y}) \leq \mathbb{E}_{z \sim Z} h(\mathbb{P}\left(Y | Z = z\right))$ follows since $\mathbb{E}_{z \sim Z} h(\mathbb{P}\left(Y | Z = z\right))$ is the error of the optimal classifier of $Y$ as a function of $Z$. We choose $\hat{Y}$ to be such an optimal classifier and thus satisfy the above inequality, with equality. Next, the second inequality in (122) holds for for *any* classifier $\hat{Y}$ derived from $Z$. The argument below is a slight generalisation of the argument in Madras et al. (2018). Define $f(z) = \mathbb{P}\left(\hat{Y} = 1 | Z = z\right)$. Note that for $a \in \{0, 1\}$, $\mathbb{P}\left(\hat{Y} = 1 | A = a\right) = \int f(z) d\mu_a(z)$. Thus

$$\mathbb{P}\left(\hat{Y} = 1 | A = 0\right) - \mathbb{P}\left(\hat{Y} = 1 | A = 1\right) = \int f(z) d\mu_0(z) - \int f(z) d\mu_1(z) \tag{123}$$

$$\leq \sup_{g | \|g\| \leq 1} \left|\int g(z) d\mu_0(z) - \int g(z) d\mu_1(z)\right| \tag{124}$$

$$= \|\mu_0 - \mu_1\|_{TV}, \tag{125}$$

where we have used $|f| \leq 1$ in the second line. Repeating the argument also for $\mathbb{P}\left(\hat{Y} = 1 | A = 1\right) - \mathbb{P}\left(\hat{Y} = 1 | A = 0\right)$, we obtain the second inequality in (122).

We now turn to the first statement, (121). Let $\hat{Y}$ be a classifier of $Y$ as a function of $(X, A)$. Observe that thus by definition $\mathbb{P}\left(\hat{Y} | X, A\right)$ induces a distribution on the set $\{0, 1\}$, and thus may be considered as a representation $Z := \hat{Y}$ of $(X, A)$ on that set. We now relate the properties of this $Z$ as a representation to the quantities $\epsilon(\hat{Y})$ and $\Delta_{SP}(\hat{Y})$. Similarly to the argument above, the first part of (121) follows since $\mathbb{E}_{z \sim Z} h(\mathbb{P}\left(Y | Z = z\right))$ is the best possible error over all classifiers. For the second part, note that since $\hat{Y}$ is binary, we have

$$\mathbb{P}\left(\hat{Y} = 1 | A = 0\right) - \mathbb{P}\left(\hat{Y} = 1 | A = 1\right) = -\mathbb{P}\left(\hat{Y} = 0 | A = 0\right) + \mathbb{P}\left(\hat{Y} = 0 | A = 1\right). \tag{126}$$

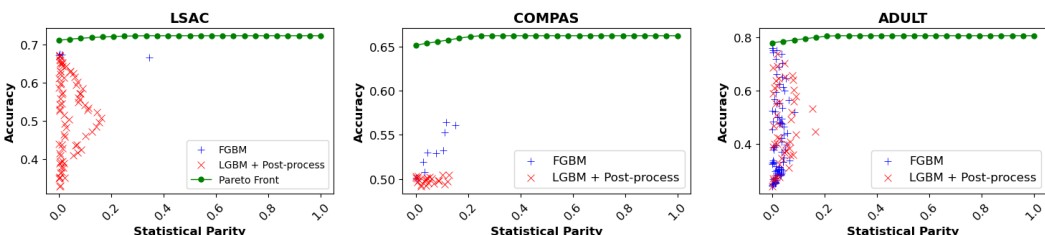

Figure 4: Comparing common fair classification pipelines to the MIFPO Pareto front, for LSAC, COMPAS, and ADULT datasets. For FGBM and LGBM+Post Process methods, each point represents a trade-off obtained at a single hyper-parameter configuration.

It follows that

$$\|\mu_0 - \mu_1\|_{TV} = \frac{1}{2} \sum_{v \in \{0,1\}} \left| \mathbb{P}\left(\hat{Y} = v | A = 0\right) - \mathbb{P}\left(\hat{Y} = v | A = 1\right) \right| \tag{127}$$

$$= \left| \mathbb{P}\left(\hat{Y} = 1 | A = 0\right) - \mathbb{P}\left(\hat{Y} = 1 | A = 1\right) \right| \tag{128}$$

$$= \Delta_{SP}(\hat{Y}), \tag{129}$$

where we have used (126) for the second to third line transition. This completes the proof of the second part of (121). □

### H.1 COMPARISON TO COMMON FAIR CLASSIFIERS

Here we use **(i)** FairGBM Cruz et al. (2023), an *in-processing*, method where a boosting trees algorithm (LightGBM) is subject to pre-defined fairness constraints, and **(ii)** Balanced-Group-Threshold Jang et al. (2021), a *post-processing* method which adjusts the threshold per group to obtain a certain fairness criterion. For FairGBM we have used the original implementation provided by the authors, while for Balanced-Group-Threshold post-processing we have used the implementations available via Aequitas, Saleiro et al. (2018), a popular bias and fairness audit toolkit.

It is important to note that, as a rule, common fairness classification methods are not designed to control the fairness-accuracy trade-off explicitly. Instead, in most cases, these methods rely on rerunning the algorithm for a wide range of hyperparameter settings, in the hope that different hyperparameters would result in different fairness-accuracy trade-off points. However, there typically is no direct known and controlled relation between hyperpatameters and the obtained fairness-accuracy trade-off. For FairGBM, we utilized the hyperparameter ranges specified in the original paper, Cruz et al. (2023). In the case of the balancing post-processing method, we conducted a grid search over the full range of all possible hyperparameters to ensure a comprehensive analysis.

Figure 4 shows the MIFPO computed Pareto front, and all hyperparameter runs of the two algorithms above, with accuracy evaluated on the test set. These experiments demonstrate the following two points: **(a)** The standard classifiers achieve a considerably lower accuracy than what is theoretically possible at a given fairness level. **(b)** the existing methods are also unable to present solutions for the full range of the statistical parity values. The values from the FGBM and the post-processing algorithms all have statistical parity $\leq 0.2$. Similarly to to the case of fair representations, these results emphasize the limitations of current fair classifiers in achieving optimal trade-offs between accuracy and fairness across the full range of fairness values.

