# OpenReview forum: "Efficient Fairness-Performance Pareto Front Computation"
_ICLR.cc/2025/Conference — Submitted to ICLR 2025_

### Official Review · Reviewer_G4EM · 2024-10-28

**Soundness:** 3
**Presentation:** 2
**Contribution:** 2
**Rating:** 5
**Confidence:** 4

**Summary:**

This paper introduces a model-independent method called MIFPO for computing the Pareto front of performance versus fairness in machine learning classification models. MIFPO achieves this by discretizing the space of possible distributions of the target variable and formulating a concave minimization problem with linear constraints. This problem can be efficiently solved using off-the-shelf solvers, such as the disciplined convex-concave programming (DCCP) framework. Experimental evaluations demonstrate MIFPO’s advantages over existing methods in both fair representation learning and fair classification.

**Strengths:**

1. **Use of Structural Properties**: The approach leverages the structural properties of optimal fair representations to efficiently approximate the Pareto front. By exploring previously underutilized properties, the authors have created a way to directly compute the fairness-performance front without a specific model structure.

2. **Model-Independent**: MIFPO’s model-agnostic nature allows it to serve as a comparative benchmark against complex fair representation models and classifiers, contributing to broader applications in fair ML model assessment.

**Weaknesses:**

1. **Generalizability Issue:** The MIFPO method relies on discretizing the target space, which limits its application to finite or low-dimensional scenarios. This assumption restricts the method's generalizability to cases where target variables are continuous or involve multi-dimensional labels. A preliminary exploration of alternative methods that could apply to these contexts, or potential theoretical adjustments, would strengthen the paper and broaden its impact.

2. **Insufficient Coverage and Comparison with Related Works:** A major concern is the limited discussion of related works, which affects the paper’s positioning and diminishes the impact of MIFPO’s contributions in the context of established fairness-performance trade-offs. This trade-off problem is well-explored in fairness in machine learning, with numerous studies proposing methods to balance these objectives. However, these methods are not cited or analyzed here. Beyond just discussion, comparing MIFPO’s empirical results against these established approaches is essential to validate its contribution, demonstrating where MIFPO achieves improvements or differs significantly in performance. While the following papers represent a sample, many other relevant studies would also merit discussion and empirical comparison:

- [1] Wang, Hao, et al. "Aleatoric and epistemic discrimination: Fundamental limits of fairness interventions." Advances in Neural Information Processing Systems 36, 2023.

- [2] Dehdashtian, Sepehr, et al. "Utility-Fairness Trade-Offs and How to Find Them." Proceedings of the IEEE/CVF Conference on Computer Vision and Pattern Recognition, 2024.

- [3] Sadeghi, Bashir, et al. "On characterizing the trade-off in invariant representation learning." Transactions on Machine Learning Research, 2022.

- [4] Menon, Aditya Krishna, and Robert C. Williamson. "The cost of fairness in binary classification." Conference on Fairness, Accountability and Transparency, PMLR, 2018.

- [5] Zhao, Han, et al. "Inherent tradeoffs in learning fair representations." Journal of Machine Learning Research, 2022.

3. **Computational Complexity and Practicality:** The MIFPO implementation relies on DCCP which can be computationally intensive, especially for large datasets. The lack of guarantees for global minima in DCCP further raises questions about MIFPO’s ability to reliably converge to optimal solutions on larger, more complex datasets. Addressing scalability concerns would improve MIFPO's applicability to real-world scenarios. Exploring alternative solvers or optimization frameworks with scalability features, or clarifying computational requirements and limitations for different dataset sizes, would make the method more accessible to practitioners.

4. **Consideration of Alternative Fairness Definitions:** The paper exclusively uses statistical parity as the fairness metric; however, multiple definitions of group fairness exist, and studies [2, 6] have shown that statistical parity may not be ideal in practice since it does not consider target attributes. Alternative metrics such as Equality of Odds and Equal Opportunity are often more practical and widely used. The paper would be significantly strengthened if the authors discussed how MIFPO could be adapted to incorporate these alternative fairness definitions, as doing so could broaden its applicability and relevance in real-world settings.

- [6] Hardt, Moritz, et al. "Equality of opportunity in supervised learning." Advances in neural information processing systems 29, 2016.

**Questions:**

All the questions and suggestions are mentioned in the Weaknesses section.

---

> ### Author Response · Authors · 2024-11-21
>
> Many thanks for your feedback!
>
>
>
> We were glad to learn that our exploitation of new structural properties of fair representations, and the model independence of the resulting algorithm, were found to be strengths of the paper.
>
>
> We now address the issues raised in the review.
>
>
> >2. Insufficient Coverage and Comparison with Related Works...
>
> Thanks again for the list of recent references!
> Please see the separate thread above were we discuss in detail the relation of our work to *all* papers suggested in this and other reviews.
>
> As mentioned there, most of the works in the list have not appeared in our original review either because they deal purely with classification rather than representations, or because they are very recent (or both). Nevertheless, we discuss how our approach is different from all other approaches. We also explain why it would be difficult to directly compare our result with some of the other represtation related work (but not FNF).
>
> In view of the discussion in the References thread:  Would the reviewer agree that our theoretical results on representations, and the associated algorithm, are new? Also, does this discussion resolve the issue of insufficient literature coverage?
>
>
> >3. Computational Complexity and Practicality: The MIFPO implementation relies on DCCP which can be computationally intensive, especially for large datasets...
>
> We believe this is a misunderstanding.
> It is an important feature of MIFPO that it *does not* depend on the size of data at all!
>
> The data is used to construct histograms. However, once we have them, the data may be discarded. Note that this is also in constrast to many other methods. For instance, in the reference [1] from this review, they have to subsample the already relatively small datasets COMPASS and Adult, in order for their code to run.
>
> Does this help resolve the computational complexity point?
>
>
> Regarding the lack of global convergence guarantees for DCCP:
> Generally, we certainly agree that exploring other solvers would be interesting. At the same time, we would like to make three points: **(a)** There exist methods that solve the convex-concave problem with guarantees, although we have not evaluated them here.
>
> **(b)**  Any front that we find is *a* lower bound, even if it is the not the optimal lower bound.
> This means that the true Pareto front must be at least as good as what we find. This makes the results useful even if they are not optimal. As long as we find better bounds than some other method, we know for sure ([*]) that this other method may be improved. And we have also observed empirically that we find better bounds than a few recent methods.
>
> **\(c\)** To the best of our knowledge, there is at the moment no other representation learning method that does provide guarantees (without making extremely strong assumptions).
>
> [*] "for sure" here mean "assuming that the probability estimator is correct". However, there are standard ways, such as cross validation, to validate such correctness. With also discuss this point further, in the context of "type (i)" errors, in responses in R2,R3.
>
>
> > 1.Generalizability Issue: The MIFPO method relies on discretizing the target space, which limits its application to finite or low-dimensional (target variable) scenarios.
>
> We agree that considering multi-class classification with many classes is an important question and have mentioned a few possible future work approaches in the Conlcusion section.
>
> We also note that discretisation does not necessarily imply limitations or high complexity. For instance, discretisation in the simplex may be performed via k-means algorithm, or some other clustering. Very often, high dimensional datsets cluster very well. And thus, our algorithm could be applied to such situations virtually with no changes.
>
> Moreover, we believe that most existing work that offers at least some thoeretical analysis is either completely based on the binary targets, or is evaluated only on binary targets. Would the reviewer agree with this statement?
> It is thus not completely fair to consider this a weakness of our specific work.

---

> > ### Comment · Reviewer_G4EM · 2024-11-25
> >
> > Thank you for your detailed response and the discussion in the References thread. I appreciate the effort in addressing my concerns. However, I believe the main issue with the lack of comparison with prior works remain unresolved. Below, I provide my feedback on the key points raised.
> >
> > > As mentioned there, most of the works in the list have not appeared in our original review either because they deal purely with classification rather than representations, or because they are very recent (or both)
> > >
> >
> > This statement is not accurate. The most recent paper I listed is from CVPR 2024 (papers were available in June), which is not considered **concurrent** work as per the [ICLR FAQ](https://iclr.cc/Conferences/2025/FAQ). Excluding this, other cited papers addressing fair representation learning were published one or two years ago. The authors’ argument that these works are excluded due to recency is therefore invalid.
> >
> > > We note that our original discussion of the fairness-accuracy Pareto front literature in the paper concentrated on the Pareto front of *representations*, since this is our main subject, while the references below also involve the tradeoffs for classifiers, which is a different, easier problem.
> > >
> >
> > I disagree with this statement. The central goal of your paper is to generate fair representations, a focus shared by the related works I cited. These prior works aim to produce a representation $Z$ invariant to sensitive attributes $A$ To evaluate these representations, they train classifiers to predict labels $\hat{Y}$, which are required for group fairness metrics such as Statistical Parity, Equalized Odds, and Equality of Opportunity.
> >
> > This evaluation methodology does not reduce to simply exploring “tradeoffs for classifiers” as the authors claim. Rather, it directly assesses the quality of the learned representations using standard fairness definitions. Hence, the distinction made by the authors between their work and prior research is not compelling.
> >
> > Furthermore, if your paper employs some metrics to evaluate representations, those same metrics can also be applied to prior works. This invalidates the justification for not including comparisons.
> >
> > **Additional Concern:**
> >
> > One of the points from my initial review (*4. Consideration of Alternative Fairness Definitions)* remains unaddressed by the authors.
> >
> >
> > Thank you again for your efforts in responding.
> >
> > In my initial review, I pointed out that some related works appeared to be missing and suggested that addressing them could strengthen the paper's evaluations and claims. However, without these comparisons, I believe the paper falls short of the acceptance threshold.

---

> > > ### Author Response · Authors · 2024-11-26
> > >
> > > Dear Reviewer G4EM, thank you for your response.
> > >
> > > We hope to address here the concerns that still remain from our discussion.
> > >
> > > >However, I believe the main issue with the lack of comparison with prior works remain unresolved. Below, I provide my feedback on the key points raised.
> > >
> > > Unfortunately, we are now not completely sure what the issues are.
> > > Could the reviewer please clarify the points below?
> > >
> > >
> > > ### Comparison of Methods
> > >
> > > The review asked for a discussion of additional literature, and we have provided a discussion of 8 papers, including the 5 in the review, emphasizing the differences in the approaches w.r.t our work.
> > >
> > > In view of this, would the reviewer agree that our approach is new? Does our discussion of the differences resolve the issue of comparison to existing approaches? (this is a question about the methods themselves, for performance comparison please see below).
> > >
> > >
> > >
> > > ### Classification vs Representations
> > >
> > >
> > > >These prior works aim to produce a representation $Z$ invariant to sensitive attributes. To evaluate these representations, they train classifiers to predict labels...
> > > >...Hence, the distinction made by the authors between their work and prior research is not compelling.
> > >
> > > Out of 5 works in the review, 2 deal purely with classification, with no relation to representations at all. These are the works [1],[4] (numbering as in the review). Do we agree on this?
> > >
> > >
> > > We note that in addition, some of the work suggested in other reviews ([6],[7], numbering as in References thread) also are purely classification. Our comments on classification thus concerned these 4 out of the 8 works.
> > >
> > >
> > > We certainly agree that [2],[3],[5] in the reveiew are dealing with representations. Appologies if this was unclear.
> > >
> > > Is the issue of distinction between classification and representation resolved?
> > >
> > >
> > > ### Empirical Comparison
> > >
> > > **(a)**
> > >
> > > In the paper we have compared our approach to FNF, which is a fairly cited ICLR 22 representations work. We also compared to two classifaction methods, implemented in popular classification frameworks.
> > >
> > > We note that none of the other representation works have done such comparisons, although its natural.
> > > As we mentioned earlier, we will also add a comparison with [1], although their existing code is quite hard to adapt.
> > >
> > > **(b)** Other representation work:
> > >
> > > One of the 3 representation papers in the review, [5], does not provide new algorithms. Thus there is nothing to compare to.
> > >
> > >
> > > Regarding [2] and [3]:
> > > >Furthermore, if your paper employs some metrics to evaluate representations, those same metrics can also be applied to prior works. This invalidates the justification for not including comparisons.
> > >
> > > We respectfully disagree. There is no simple/canonical way to compute Total Variation distances between two *point sets*. This in general requires either density estimation (like FNF), which is non-trivial, or indirect bounds, like our method.
> > >
> > >
> > > Would the reviewer agree that Total Variation is a natural distance for Fairness purposes?
> > >
> > > Would the reviewer agree that there is no standard easy way (*) to estmate the Total Variation between two point sets?
> > >
> > >
> > > (*) A way that does not require a paper by itself.
> > >
> > > ### Additional Notes
> > >
> > > We would like to add that we did not intend to imply that papers published in June (3 months before our submission deadline) should not be added to Related Work. They should and they will be added. Nevertheless, June is very recent, and we believe latest NeurIPS (ref [1]) is also fairly recent.

---

> ### Comment · Area_Chair_FVVL · 2024-11-24
>
> Comment: Dear Reviewer G4EM,
>
> The author discussion phase will be ending soon. The authors have provided detailed responses. Could you please reply to the authors with whether they have addressed your concern and whether you will keep or modify your assessment on this submission?
>
> Thanks.
>
> Area Chair

---

### Official Review · Reviewer_mT9S · 2024-10-29

**Soundness:** 2
**Presentation:** 2
**Contribution:** 2
**Rating:** 5
**Confidence:** 3

**Summary:**

This paper addresses the problem of understanding the critical tradeoff between fairness and accuracy. They propose strategies to obtain the Pareto frontier without performing any additional model training but instead by formulating an optimization problem.
They consider the TV as their measure of fairness, and then go on to show how for every representation, one can find an invertible representation of the same data that satisfies at least as good a fairness constraint, and has at least as good performance as the original (shown in Theorem 4.1). Next, using this result they construct a model independent optimization problem -- that they call MIFPO -- which can approximate the Pareto Front of arbitrary high dimensional data distributions, but is much simpler to solve than direct representation learning for such distribution. They specifically use off-the-shelf concave-convex programming methods to obtain the Pareto frontier. Experiments have been performed on a few benchmark datasets such as LSAC, COMPAS, and ADULT.

**Strengths:**

-- Interesting optimization-based strategy to obtain Pareto frontiers of the fairness-performance tradeoff
-- Bypasses expensive model training
-- Strength lies in being able to restrict the problem space to invertible transformations, and then use off-the-shelf concave-convex optimization strategies

**Weaknesses:**

--The presentation of the paper can be significantly improved. It is difficult to understand or properly check everything. For every subsection, it would be good to start with the main theorem/lemma/assertion being made in that subsection, then provide its significance towards arriving at your main result, and then provide the proof sketch/intuition behind it.
Specific Example: The Invertibility Theorem, The Factorisation, The MIFPO  -- all these sections start with "This section provides ...." which is sometimes already understood from the title but what is important is to mention what role this particular aspect plays in the final optimization before getting into the details.

-- In the introduction, it would be nice to summarize the main contributions as a list.
Specific Example: It would be great to have a flow-chart summarizing what are the various Theorems/Assertions made and specifically what role they play/why they are needed for the final optimization.

-- There are related works in accuracy-fairness literature which have proposed model-agnostic optimization formulations to characterize the trade-off. These two prior works share close similarities with the formulation in this paper and should be discussed.
Briefly explaining the key similarities and differences: Interestingly, while these prior works finally arrived at a convex optimization problem, but this paper arrives at a concave minimization problem with convex constraints which presents an intriguing nuance.
[1] Kim, Joon Sik, Jiahao Chen, and Ameet Talwalkar. "FACT: A diagnostic for group fairness trade-offs." International Conference on Machine Learning (ICML) 2020.
[2] Hamman, Faisal, and Sanghamitra Dutta. "Demystifying Local & Global Fairness Trade-offs in Federated Learning Using Partial Information Decomposition." International Conference on Learning Representations (ICLR) 2024.

-- There are several other related works on accuracy-fairness tradeoffs which have been missed. See the references and follow-up works from these two papers above to find more references.

**Questions:**

-- Could you discuss the implications of using different accuracy measures on your optimization formulation? Entropy accuracy seems to be challenging and also more involved than regular accuracy of prediction=true label.
-- Would the Pareto frontier optimization problem be simpler if other accuracy measures were used?
-- Could you simplify the flow chart of the mathematical arguments leading onto your main optimization (which btw has not been written together  with all the constraints)? For instance, what role does the factorisation, invertibility, etc. individually play to simplify and get to your final version?
-- What is the role of post-processing strategies?

---

> ### Author Response · Authors · 2024-11-21
>
> Thank you for your feedback!
>
> We were glad to learn that our overall strategy was found interesting, and that among strengths listed were *(a)* the bypassing of expensive model training and *(b)* the introduction of the restriction of the candidate space, to the set of invertible respresentations.
>
>
> We now turn to the issues and questions appearing in the review. Please let us know wether the discussion below addresses the concerns/questions raised in the review. We would be glad to discuss any point  further if required.
>
>
> >There are related works in accuracy-fairness literature which have proposed model-agnostic optimization formulations to characterize the trade-off. ...
>
>
> Thank you for providing the refernces!
> Please see the separate References thread above, were we discuss the relation of MIFPO with 8 papers appearing in the reviews, including the papers suggested here.
>
> Do the comments in that thread clarify the position of our work within the field and resolve the concern with related works? In particular, would the reviewer agree that the theory and algorithms presented in our work are new?
>
>
> >Interestingly, while these prior works finally arrived at a convex optimization problem, but this paper arrives at a concave minimization problem with convex constraints which presents an intriguing nuance.
>
> We assume that the intention here is that "FACT:..." paper referenced in this review has a convex objective. While their objective is indeed convex, it depends on data very weakly and provides weak bounds in the Pareto front. This was recently improved in the paper "Aleatoric and epistemic discrimination:..." ([1] in References thread above). However, their overall algorithm is no longer a standard convex optimisation. See also the items [6] and [1] in the References thread.
> Does this clarify the convexity issue?
>
>
> >Could you discuss the implications of using different accuracy measures on your optimization formulation? Entropy accuracy seems to be challenging and also more involved than regular accuracy of prediction=true label.
>
> Our approach in general can handle any concave function $h$ (lines 172-186) to measure the preservation of information about $Y$. The DCCP framework specifically places some restrictions on what $h$ may be implemented, but allows  both the accuracy (eq (4)) and the entropy function.
> In the experiments, we only use the accuracy function for comparison purposes, both for representations, and classification, because all other work only uses the accuracy.
>
>
> >What is the role of post-processing strategies?
>
> We assume the intention here is to the post-procesing strategies that we use as a benchmark in the Experiments section.
>
> We discuss this post processing briefly on lines 483-502 in the paper.
>
> In more detail:
> Post processing is an approach to obtaining fair classifiers that involves first training some (non-fair) classifier, and then modifying its output in ceratain way, to obtain fair classifiers. We use it as one of the two benchmark classification methods, against which we compare MIFPO.
>
>
> Continued below

---

> > ### Author Response · Authors · 2024-11-21
> >
> > >Could you simplify the flow chart of the mathematical arguments leading onto your main optimization (which btw has not been written together with all the constraints)? For instance, what role does the factorisation, invertibility, etc. individually play to simplify and get to your final version?
> >
> >
> > We have some discussion and a high level overview of the different components of the work in the Introduction, line 80-102. We believe the main optimisation problem is fully presented in Definition 5.1 and the overall algorithm described in Algorithm 1.
> >
> >
> > However, we agree that the exposition is somewhat condensed, mainly due to space concerns. We will now attempt to clarify the situation, and will add further clarifications in the final version of the paper.
> >
> >
> > Perhaps the best way to describe the relation of different parts of the work is to start with Algorithm 1. This describes the actual steps: Learning probabiliy estimators from data, building the histograms, and then constructing the MIFPO problem instance and solving it with a solver.
> >
> > The invertability and factorisation results serve as theoretical motivation and justification as to why this particular process should provide the actual true Pareto front of the data.
> >
> > Factorisation says that instead of considering all features $x$, one can consider only the distributions $P(Y|x)$, since two $x$ with same such distrbution are identical for Pareto front purposes.
> > This reduces the object that we need to represent from a distribution on $R^d$ to a distribution on $\Delta_Y$. And $\Delta_Y$ is a much smaller space, which is easier to discretise.
> >
> > This effectively reduces the size of the "source space" without losing accuracy.
> >
> > The Invertibility Theorem, on the other hand, allows us to control the size of the "target space" of the representation. That is, how many points $z$ in the representation do we need to have in order not to loose accuracy (some additional details on this are on lines 333-342).
> >
> > Together, these two results determine how  many variables the MIFPO instance would have.
> >
> > Does this help clarify the overall perspective on the paper?

---

> > > ### Comment · Reviewer_mT9S · 2024-11-25
> > > **Response**
> > >
> > > Thank you for your responses! I will take them into account in arriving at my final decision.
> > > The paper doesn't seem to be updated (or could the edits be colored?) It would have been nice to have been able to review an updated version of the paper with some of the planned edits already incorporated, particularly the suggestion on a simplified flow-chart and organization/presentation of the results in a more accessible manner.

---

> > > > ### Author Response · Authors · 2024-11-26
> > > > **Response regarding the new version.**
> > > >
> > > > Thank you for your response!
> > > >
> > > > We have incorporated many of the discussions suggested in an updated version of the paper, which is now available. To accommodate these additions while maintaining the page limit, we have relocated most of the classification-related evaluations to the supplementary material.
> > > >
> > > > ### Please note that this is not the final version, and we plan to further revise the paper to incorporate all feedbacks from all reviewers in the final version.

---

> ### Comment · Area_Chair_FVVL · 2024-11-24
>
> Comment: Dear Reviewer mT9S,
>
> The author discussion phase will be ending soon. The authors have provided detailed responses. Could you please reply to the authors with whether they have addressed your concern and whether you will keep or modify your assessment on this submission?
>
> Thanks.
>
> Area Chair

---

### Official Review · Reviewer_YRP9 · 2024-10-31

**Soundness:** 3
**Presentation:** 3
**Contribution:** 3
**Rating:** 8
**Confidence:** 4

**Summary:**

This paper presents a method for calculating the Fairness-Accuracy Pareto front for all fair representations for a particular fairness-constrained learning task without needing to explicitly train representation models. As such, this front describes a space of possible trade-offs for fair representations in terms of their fairness vs. accuracy, which might have downstream applications in benchmarking particular trained fair representations and check if their performance still has room for improvement. The authors accomplish this through transforming the fair representation learning problem to a much lower-dimensional problem through observing a mapping from the input space to the simplex of the output space, $\Delta_{\mathcal{Y}}$. In binary classification, this is particularly appealing because $\Delta_{\mathcal{Y}} = [0, 1]$, which is nicely discretizable and much "more compact" than the arbitrary space of inputs. This results in a concave *minimization* program that, although unsolvable through nice black-box methods for convex optimization (it is concave *minimization* not *maximization*), still has easily estimable quantities from data and is experimentally feasible through certain DCCP frameworks. The authors show this in their suite of experiments over various standard fairness datasets.

**Strengths:**

Overall, I found the main idea of this paper and the technical contribution very nice, and the authors mostly present their results in a well-organized way. I am not an expert in the fair representations literature, so I am not sure how these methods are positioned with respect to other fairness-ensuring methods (i.e. inprocessing methods, postprocessing methods), so I am unclear whether the paper as a whole is well-motivated (it very well may be, I could just be ignorant of how widely adopted fair representation learning is). But, if we take it as given that it is desirable to construct new ways to benchmark fair representations without needing to train them explicitly with complicated nonconvex NNs and such, then this paper presents a worthwhile and interesting contribution.

**Originality.** As far as my knowledge goes, this paper is an original contribution to the fair representation learning literature. It is particularly novel because it presents a way to compute a Pareto frontier for fairness-accuracy tradeoffs without explicitly training a representation itself. The authors come to this computation through analyzing the theoretical properties of a fair representation by first positing an intuitive optimization problem and then breaking down its constituent parts into quantities that are easily estimable from the data. The key step in doing so is interesting and, in my opinion, novel --- the authors reduce the search space of the problem from the input space (which can be any arbitrary subset of $\mathbb{R}^d$) to $\Delta_{\mathcal{Y}}$ through basic properties of representations.

**Quality.** The result itself is interesting and significant, and I particularly believe that the general technique of reducing the search space to $\Delta_{\mathcal{Y}}$ to make the computation feasible is interesting. The experiments seem to be comprehensive, and Algorithm 1, the implementation to solve MIFPO, relies on simple primitives that are easily estimable from the dataset. The derivation in Sections 3, 4, and 5 are sound as far as my reading of the results go, though I did not carefully step through the proofs of the main theorems and lemmas.

**Clarity.** The writing itself is mostly clear, although there are some notation changes I would suggest to make the exposition clearer (see "Weaknesses"). The main suggestion would be to stick with probability notation (e.g. $\mathrm{Pr}(Z = z \mid A = a)$, etc.) throughout the paper or to stick with the defined notation (with $\alpha, \beta, \rho, \mu$, etc.) throughout the paper. Switching between the two obscures some of the results and made reading the results slightly more difficult. Besides that, the paper itself is well-structured and, overall, gives a clear exposition of the main technique. Besides these notational difficulties, I had little trouble following along.

**Significance.** Because I am not an expert or well-versed in the specific fair representation subfield of the fairness literature, I cannot speak to the motivation of this particular method and whether it is a big gap in the literature. However, if it *is* desirable to have methods for computing such Pareto frontiers without explicit training of representations in order to benchmark fair representations well, I believe this is a significant contribution.

**Weaknesses:**

There are a a couple of weaknesses I found in the paper, but none are significant enough to warrant rejecting the paper, in my view. Most of the weaknesses relate to the clarity of the paper, but the main technical weakness, in my eyes, is that the resulting optimization problem, though simpler, still relies on nonprovable guarantees --- it is a concave minimization problem, so we have no guarantees of global optima.

- **Concave minimization.** The authors themselves acknowledge in the conclusion that this is a weakness of their work, but it would be nice to see some more exposition justifying that computing this optimization problem, though it does not provably converge to global optima, is still a good thing to do in the context of constructing this Pareto front (more in "Questions").
- **Figures are slightly confusing.** I found the figures to be slightly confusing, as the notation in the figures doesn't quite match up with the notation in the paper, and the figures themselves are blurry and don't quite give a clear picture to what's going on, in my opinion. I would suggest going a bit lighter on the notation in the figures to make them a bit more comprehensible; for example, I don't think that Figure 2(b) needs the $\overline{T}$ full definition in it, and I don't think that Figure 2(a) needs a presentation of the probability distribution on $v, u, u'$ in the form of the squiggly red line and the accompanying $\beta$ labels. Cleaning up the figures to be more presentable would help in the clarity of presentation.
- **Notational issues.** My main issue with the clarity of the paper is that there is much switching back and forth between established notation and introducing of new notation. I think the paper would be clearer if the authors either stuck to just using the probability notation through, so it is clear what probabilities factorize into what (i.e. sticking to just presenting $T_a(z, s)$ instead as just $P(Z = z \mid X = s, A = a)$), or at least introducing notation that is a bit more evocative of what each quantity is. I would suggest perhaps using notation that invokes the relationship between $A, Z, X,$ and $Y$ in the symbols themselves. I found that Sections 4 and 5 were a bit heavy on introducing new notation and then not using it everywhere, such as in the definition of the $r$ variable in Section 5.2. I was still able to track the main arguments, but I believe that clarifying the notation a bit would assist in the overall clarity.
- **A couple of typographical issues/nitpicks.** These don't have an impact on my overall impression of the paper, but I just wanted to point out some small nitpicks I noticed to assist the authors in revising:
- "MIFPO" is never defined as an acronym. I assume that the acronym should stand for "Model Independent Pareto Front Optimization," but I guess it would be MIPFO then?
- In Equation 5, as a minor clarity fix, I would treat make the dependence of $Z$ on $\theta$ explicit and then just mention that, for the rest of the paper, we can assume that it is implicit in the distribution governing $Z$.
- In Section 5.2, in the first paragraph, I would suggest writing the Lemma D.1 more formally in the main body, or, at least, explaining the $k$ point approximation a bit more. I found it a bit confusing until I checked the Appendix section.

**Questions:**

My main questions concern how the authors view the nonconvexity of the entire optimization problem as a major or minor issue. They point out in the Conclusion section that their optimization problem, as it is concave *minimization*, does not provably converge to global optima, but I wasn't sure after reading the paper why solving the MIFPO program with DCCP is satisfactory. We see that the fairness-performance curve in Section 6 leaves some points that FNF doesn't quite reach, but is there anything we can formally say about how close to true optimal this fairness-performance curve is? I think that would rely on formal guarantees for DCCP, but I am unfamiliar with such guarantees.

Related to this question: how do we evaluate how good this "local" optimal Pareto frontier is? And what modifications to the MIFPO program seem promising to make it provably convex? My intuition suggests that the problem lies in the choice of $h$, but, in that case, it seems there's not much one can do.

---

> ### Author Response · Authors · 2024-11-21
>
> Thank you very much for the detailed review, and for the favourable score!
>
> We were very glad to learn that the review found our results novel, significant, and well presented. Thank you!
>
>
> We note that in a separate References thread, we discuss the relation of work with 8 recent papers,  mentioned in other reviews. This may further help place our work in the context of existing results.
>
>
> We now address the questions in the review.
>
> > how the authors view the nonconvexity of the entire optimization problem as a major or minor issue.
>
> Before discussing the sources of error, we would like to note that in principle,
> our results are lower bounds. Meaning that the true Pareto front must be at least as good as what we find. This makes the results useful even if they are not optimal. As long as we find better bounds than some other method, we know for sure ([*]) that this other method may be improved. And we have also observed empirically that we find better bounds than a few recent methods.
>
>
> Next, with respect to the true Pareto front, there are essentially 3 sources of possible error: *(i)* The error in learning the probability estimators from the data *(ii)* Discretisation errors, in discretising the histogram and choosing the representation space size (essentially the value $k$, lines 333-342). and *(iii)* the possibility that DCCP will get stuck in a local optimum.
>
>
> The error *(i)* is essentally unavoidable in any approach as long as finite data samples are invloved. Compared to other approaches, we believe we have reduced the dependence on the data to the minimum possible.  Since this is the classical ML issue, there exist a variety of ways to guarantee good performance of the probabiliy estimator, such as cross validation or sample complexity bounds.
>
> The type *(ii)* errors are controllable in principle: We can control the errors from discretisation, especially when we discretise a simple space, such as the unit interval $[0,1]$. And for the choice of $k$, we have easily applicable bounds that relate the size of $k$ to the worst possible error in the front (Supplementary Material Section D).
>
>
> Finally, type *(iii)* error: The DCCP is known to work fairly well. The reason have chosen DCCP for our work is simply since it had the most convenient Python interface and was easy to run.
> However, if local optima ever become an issue, as we mention briefly in the Conclusion, there exist
> so called branch-and-bound families of methods that *guarantee* the convergence to global optimum. These methods may have  slow run time in the worst case, but the field of this kind of optimisation as whole is in priciple well developed.
>
>
>
> [*] "for sure" here means that we assume we did not have errors in the probablity estimator. However, there are ways to control errors of the estimator, as discussed in type *(i)* errors above.
>
>
> >how do we evaluate how good this "local" optimal Pareto frontier is? And what modifications to the MIFPO program seem promising to make it provably convex?
>
> As mentioned above, every "local" front that we find is a lower bound and thus useful in priciple.
>
> We also mention that at the particular value $\gamma = 0$, i.e. perfect fairness, the MIFPO instance becomes a *linear programming* problem, equivalent to certain Optimal Transportation problem (lines 96-102, and Supplementary B). Linear programs can also be solved exactly by standard algorithms, theoretically in polynomial time. This could in priciple serve as another sanity check for goodness of quality of the solutions.
>
>
> However, in general, we unfortunately do not know how one can convert the problem to be convex. Our intuition is that this may be impossible for this problem.
>
>
>
>
>
> We will also take all the other notes in the review (figures, notation) into account in the final version of the paper.

---

> > ### Comment · Reviewer_YRP9 · 2024-11-25
> >
> > Thank you for clarifying the utility of the solutions provided in the paper! This clarifies things for me a bit, and I will keep my score.

---

### Official Review · Reviewer_Aqxr · 2024-11-01

**Soundness:** 3
**Presentation:** 2
**Contribution:** 3
**Rating:** 6
**Confidence:** 3

**Summary:**

The authors work with approximately fair representation learning, where the goal is to learn a fair representation random variable Z in $R^{d’}$. They define a $\gamma$-Fair representation as one which has a small L1 distance between the random variable Z conditioned on either a = 1 or a = 0 (where a is the sensitive attribute realization). Perfectly fair ($\gamma=0$) representation implies demographic parity / equalized odds w.r.t. attribute A for any classifier built on top of Z. The goal is to find a representation Z which minimizes total error (any concave function $h$) of the label random variable $Y$ conditioned on the representation $Z$. That is, we want to be able to reconstruct the ground truth conditional label distribution $Y$ with only the ability to modify the representation $Z$, subject to the constraint that $Z$ is $\gamma$-fair.

The main issue tackled by the paper is the fact that it can be difficult to work in the representation space $R^{d’}$ if the representations are allowed to be arbitrarily complex. Therefore, the authors focus on _invertible_ representations (line 240), which are those with a small basis in the original feature space (X,A). In theorem 4.1, the authors demonstrate that at least w.r.t. the fairness constraint, it suffices to search only in the space of invertible representations, whose total error is guaranteed to be “good enough”.

Next, in section 5, the authors show how to use the invertibility theorem 4.1 in order to efficiently optimize over the representation space. Finally, in Section 6, the authors experiment with the proposed method, MIFPO, and compare against an existing fair representation learning algorithm FNF. They demonstrate that by estimating the base conditional probabilities with calibrated classifiers, MIFPO allows for construction of fair classifiers which tradeoff fairness and accuracy to a specified degree better than prior methods.

**Strengths:**

I really enjoyed the fact that we can restrict to looking at a small, finite number of “parents” of points in the representation space, and this totally describes the set of pareto efficient transformations (I have a soft spot for such factorizations). The experiments also seem reasonable, and tie in with the theory directly. The fact that the proposed method MIFPO utilizes existing solvers is also a strength, given the power of these existing solvers in practice. Overall, the paper seems to propose a clear improvement over the compared method from previous work (FNF).

**Weaknesses:**

I have not worked in and am not familiar with the area of fair representation learning, and did not check any proofs of the paper. I think a major weakness of the paper --- as someone in algorithmic fairness with some background in theory --- is that it can be difficult to read. In particular, the main body is quite notation heavy, especially as you push into section 5. For example, some of the triple indexing in equation (17) seems a bit unnecessary, given that only two orders ever appear — $r_{u,v,j}$ or $r_{v, u, j}$ — and these two are very difficult to tell apart. Perhaps having a notation guide somewhere (maybe in the appendix) can be useful, since I found myself going back and forth needing to look up the definition of each variable.

I also think that there is some room for improvement in terms of detailing what the sources of randomness are (e.g. via the parameterization $\theta$), or better describing some of the notation (see minor comments for detailed examples). This could be a consequence of my background and being unfamiliar with working with labels and representations as random variables directly.

I am open to raising my score if the readability can be improved for the general theory / fairness audience. Perhaps this will require making parts of the discussion in the main body less formal in order to better give some intuition. It may also require having more statements / sections which describe the contributions and importance of different theorems and parts at a high level (especially for Section 5.2). This can be difficult, given that you would of course like to formally spell out the concave program that you are solving in the main body. I am open to suggestions on how the readability of 5.2 can be improved.

**Questions:**

As far as I can tell, all non-pareto efficiency in the experiments come from the following factors: 1) Not enough samples 2) suboptimality of the DCCP solvers 3) Not enough expressibility of the chosen (proxy) representation space via choice of $k$ value. I believe that these should be discussed in a self-contained section within the experiments section. In addition, could there be more sources of errors which combine to make the curves not only have “pareto optimal” points in Figure 3?

Other minor comments:

1. Some inconsistent usage of \citet vs. \citep (for example in the first paragraph of the paper). Please see the reference: https://towson.libguides.com/mlastyle/in-text
2. Figure 1 seems to be slightly vertically squashed, which makes it a bit harder to read the equations. I would also recommend using a vector graphic (pdf) figure instead of a png / jpg, as the latter does not allow for highlighting / reduces accessibility. A pdf may also scale up / down better in terms of figure quality.
3. What is figure 1 depicting? Can you provide an explanation in the caption?
4. What is the randomness over in $P(Z|X = x, A=a)$ in line 78? Is there assumed to be an underlying distribution $D$ on $X \times Y$? Note: I believe the randomness is over theta, the parameters of the representation, but it may be helpful clarify this earlier if using the notation $P(Z|X = x, A=a)$ in the introduction.
5. Is $P(Y|X=x)$ the ground truth conditional label distribution (line 174)? If so, is $P(Y|Z=z)$ the same ground truth conditional label distribution in the representation space Z? I guess I am just a bit confused. I thought we defined Y as a random variable of X,A, how can we condition Y on a representation z?
6. To make sure I understand footnote 1, if we consider $\beta_a(s)$ in line 228, this should formally be written as $P(X=s|A=a) = P(X = (x, a) | A = a)$, where in the latter, the two realizations of $a$ are identical?
7. Invertible representation seems to be a key concept, maybe it should be a formal definition in line 240?
8. The choice of $k$ seems quite important for MIFPO. Do you have any ablations of the method performance across $k$ size? I’m assuming the computational complexity scales in $k$?
9. How does the work relate to Impossibility results for fair representations (Tosca Lechner, Shai Ben-David, Sushant Agarwal, Nivasini Anathakrishnan)? Is it that they require perfect statistical parity (i.e., TV distance 0)?
10. Missing citations for “Health”, “Crime”, and “Law” datasets. What are these datasets?
11. Missing citations for Adult, LSAC, and COMPAS datasets.
12. More generally, more details are needed for the experiments, how many train points for the calibrated classifiers? Do you hold out some fixed percent for post-processing?
13. “Pareto front is flat for Law dataset” (468-473), but the MIFPO curve doesn’t seem very flat in Figure 3a? In fact, the vertical variability looks similar to the health dataset.
14. Non-monotonicity of pareto curve in Figure 3a: is this only due to sample size issues?
15. Figure 4: inconsistent legends between plots. Could probably remove the legends from COMPAS and ADULT plots, or ensure that all legends are identical. Pareto Front should also maybe be labeled as “Pareto Front as Found by MIFPO (Ours)”, or something similar for clarity.
16. As a reader, I gained very little (if anything) from Figure 1 and 2. Ideally, they should be self-contained, or at least have an explanation somewhere in the text. Neither figure is explained in the main text or the caption.
17. In the appendix align environments, consider using \nonumber in order to hide the equation numbers of unreferenced lines. For example, equation 118 is — to my knowledge — not referenced anywhere.
18. The full concave program should be stated somewhere, instead of for example in Def. 5.1 saying that we minimize equation (17) subject to constraints (15-16) and (20-22) which are spread throughout the paper.

---

> ### Author Response · Authors · 2024-11-21
>
> Thank you for the very detailed review!
>
> We were very glad to learn that the reviewer enjoyed our invertibility and factorisation results. One does not read such things everyday, thank you!
> We were also very glad to see that the paper was overall found to be a clear improvement over FNF.
>
>
> In what follows we address the questions and issues raised in the review.
> Please let us know whether our responses indeed resolve the issues. We will be glad to discuss any of the points further.
>
>
>
> >all non-pareto efficiency in the experiments come from the following factors: 1) Not enough samples 2) suboptimality of the DCCP solvers 3) Not enough expressibility of the chosen (proxy) representation space via choice of $k$ value. I believe that these should be discussed in a self-contained section within the experiments section
>
> Indeed, these are exactly all three possible sources of error. We now discuss these sources in more detail. A similar discussion was also included in our response to R3. The discussion will be included in the paper.
>
> Before discussing the sources of error, we would like to note that in principle,
> our results are lower bounds. Meaning that the true Pareto front must be at least as good as what we find. This makes the results useful even if they are not optimal. As long as we find better bounds than some other method, we know for sure ([*]) that this other method may be improved. And we have also observed empirically that we find better bounds than a few recent methods.
>
>
> Next, with respect to the true Pareto front, there are essentially 3 sources of possible error: *(i)* The error in learning the probability estimators from the data  *(ii)* Discretisation errors, in discretising the histogram and choosing the representation space size (essentially the value $k$, lines 333-342). and *(iii)* the possibility that DCCP will get stuck in a local optimum.
>
>
> The error *(i)* is essentally unavoidable in any approach as long as finite data samples are invloved. Compared to other approaches, we believe we have reduced the dependence on the data to the minimum possible.  Since this is the classical ML issue, there exist a variety of ways to guarantee good performance of the probabiliy estimator, such as cross validation or sample complexity bounds.
>
> The type *(ii)* errors are controllable in principle: We can control the errors from discretisation, especially when we discretise a simple space, such as the unit interval $[0,1]$. And for the choice of $k$, we have easily applicable bounds that relate the size of $k$ to the worst possible error in the front (Supplementary Material Section D).
>
>
> Finally, type *(iii)* error: The DCCP is known to work fairly well. The reason have chosen DCCP for our work is simply since it had the most convenient Python interface and was easy to run.
> However, if local optima ever become an issue, as we mention briefly in the Conclusion, there exist
> so called branch-and-bound families of methods that *guarantee* the convergence to global optimum. These methods may have  slow run time in the worst case, but the field of this kind of optimisation as whole is in priciple well developed.
>
>
> [*] "for sure" here means that we assume we did not have errors in the probablity estimator. However, there are ways to control errors of the estimator, as discussed in type *(i)* errors above.
>
> Continued below

---

> > ### Author Response · Authors · 2024-11-21
> >
> > ## Readability
> >
> > > having a notation guide somewhere (maybe in the appendix) can be useful,
> >
> > Thank you for the suggestion, a notation guide will be added in the Supplementary.
> >
> > We are also not quite happy with the $r_{u,v,j}$ notation and we consider how it may be improved. The situation that this notation has to describe is rather involved. We hoped that Figure 2a somewhat clarifies this notation.
> >
> > >I also think that there is some room for improvement in terms of detailing what the sources of randomness are.
> >
> > Please see our responses to the "Minor Comments" 3,4,5 below. Please let us know
> > wether this clarifies the situation with the sources of randomness.
> >
> > The Problem Setting (Section 3), will be expanded to include the above clarifications.
> >
> >
> > >Perhaps this will require making parts of the discussion in the main body less formal in order to better give some intuition. It may also require having more statements / sections which describe the contributions and importance of different theorems and parts at a high level. (especially for Section 5.2).
> >
> > We plan to add the following overview of the different components. We will also add a higher level descriptoin of Section 5.2 specifically.
> >
> > **Hilgh level relation of different components (overview):**
> >
> > Perhaps the best way to describe the relation of different parts of the work is to start with Algorithm 1. This describes the actual steps: Learning probabiliy estimators from data, building the histograms, and then constructing the MIFPO problem instance and solving it with a solver.
> >
> > The invertability and factorisation results serve as theoretical motivation and justification as to why this particular process should provide the actual true Pareto front of the data.
> >
> > Factorisation says that instead of considering all features $x$, one can consider only the distributions $P(Y|x)$, since two $x$ with same such distrbution are identical for Pareto front purposes.
> > This reduces the object that we need to represent from a distribution on $R^d$ to  a distribution on $\Delta_Y$. And $\Delta_Y$ is a much smaller space, which is easier to discretise.
> >
> > This effectively reduces the size of the "source space" without losing accuracy.
> >
> > The Invertibility Theorem, on the other hand, allows us to control the size of the "target space" of the representation. That is, how many points $z$ in the representation do we need to have in order not to loose accuracy (some additional details on this are on lines 333-342).
> >
> > Together, these two results determine how  many variables the MIFPO istance would have.
> >
> >
> >
> >
> >
> > ## Minor Comments
> > Due to time constraints, we address some of the Minor Comments in the review.
> > Please let us know if you'd like us to address any other partciular comment.
> > All of the comments will be taken into account in the final version of the paper.
> >
> >
> > >4. What is the randomness over in $P(Z|X = x, A=a)$ in line 78?
> >
> > The representation is defined to be a *random map*, rather than a deterministic map. So if we want to represent $(x,a)$, we should sample from $P(Z|X = x, A=a)$.
> > Alternativey, one can say that $(x,a)$ is represented by a distribution on $z$, rather than by a single $z$.
> > So the randomness is just representation's own intrinsic randomness.
> > This is similar to how one, for instance, allows transportation plans, rather than transportation maps, in optimal transportation.
> > Considering random maps is fairly standard in the literature.
> >
> >
> > >3. What is figure 1 depicting? Can you provide an explanation in the caption?
> >
> > Figure 1 intends to depict the general setup of fair representations, with the key quantities illustrated.
> > $Z$ in the middle is the representation space. The two spaces on the sides are the features $X$ restricted to $A=0$ and $A=1$ respectively.
> > It is shown that each $z$ can have multiple sources from each side (prior to invertibility theorem..). Most importantly, it is shown that $P(Y|Z=z)$,
> > the information about $Y$ at the representation point $z$, is decomposed into $P(Y|x,a)$ at $z$'s parents $x,a$.
> >
> > >5. Is $P(Y|X=x)$ the ground truth conditional label distribution (line 174)? If so, is $P(Y|Z=z)$ the same ground truth conditional label distribution in the representation space Z? I guess I am just a bit confused. I thought we defined Y as a random variable of X,A, how can we condition Y on a representation z?
> >
> > This is related to the previous point.
> > The variable $Z$ is coupled with $X,A$. That is, we have defined some representation $P(Z|X = x, A=a)$. With this representation, we can then also ask the inverse question: given $Z$, what do we know about $X,A$? And then,  this in turn gives us information about $Y$. This is roughly the content of the formula for $P(Y|Z)$ in the bottom of Figure 1. Rigorously, the formula itself is a consequence of the conditional independence (eq. (1)), which defines the representations.
> >
> >
> > Continued below

---

> > > ### Author Response · Authors · 2024-11-21
> > >
> > > >9. How does the work relate to Impossibility results for fair representations (Tosca Lechner, Shai Ben-David, Sushant Agarwal, Nivasini Anathakrishnan)?
> > >
> > > Our understanding is that this paper cosiders some issues with Fair Representations under *distribution shifts*. It does not deal directly with the accuracy-fariness tradeoffs, and does not provide algorithm for evaluating these quantities. In our work, we do not consider distribution shifts.
> > >
> > > We note that in the separate References thread, in addition to the comments above, we have also discussed  7 other papers suggested in the reviews.

---

> > > > ### Comment · Reviewer_Aqxr · 2024-11-21
> > > >
> > > > Thanks for the detailed responses; sorry for giving too many questions! Most are minor anyways. I mostly agree with reviewer YRP9 that this is a nice paper and will therefore raise my score by a point. I think a lot of the minor issues can be addressed in the next version (such as notation, figures / captions, and explanation of randomness a la the above discussion). I look forward to reading the next draft of the paper!

---

### Official Review · Reviewer_a4f1 · 2024-11-04

**Soundness:** 3
**Presentation:** 3
**Contribution:** 2
**Rating:** 6
**Confidence:** 2

**Summary:**

This paper has two main technical contributions: The first is to deeply analyze structural properties of optimal fair representations, and the other contribution is develop a method to compute the Pareto front for data distributions with any dimension. A particular advantage of this new method is its model independence: By using well-established optimization techniques, the computation of such Pareto front does not require an extensive of training of representation models; instead, much simpler data operations shall be sufficient.

**Strengths:**

1.This paper studies an interesting and important problem, as realizing the Pareto front is indeed important for any practical problems.

2. This paper is theoretically rigorous. The main techniques to achieve the goal, such as reduction to focus on invertible representations and solve the optimization problem (17) with restrictions listed below, are sound with formal justifications. Consequently, the method is theoretically credible.

**Weaknesses:**

The sensitive attributes here are assumed to be binary, but recently there have been some results for more general cases (multi-class, multi-group), and computing Pareto front for those cases are certainly more desirable, yet the setting in this paper is limited.

**Questions:**

I do not have any significant questions.

---

> ### Author Response · Authors · 2024-11-21
>
> Thank you for the feedback and for the favourable review!
>
> We were glad to learn that the problem the paper addresses was found interesting and the approach theoretically credible.
>
>
> We certainly agree that treating non-binary sensitive attributes is practically important. Moreover, we believe the extension of our approach to multiple values can be done fairly easily. For instance, the Invertibility Theorem holds even when $|A|>2$. (that is , each point $z$ would have only two parents, even if $A$ can take more than two values).
>
>
> That said, we have based our setup on earlier, fairly well cited work (Balunovic et al, ICLR 22 (FNF)) which also uses binary sensitive attributes.
> We also note that many works that allow non-binary attributes (even continuous ones), use somewhat less sensitvie fairness measures. For instance, the pure mutual information $I(Z,A)$ (Song et al 2019, AISTATS 2019), or the RKHS based independece measures (ex. references [2],[3] in our References thread above), are influenced very weakly by small groups (say if $P(A=0)$ is small), and thus would be difficult to analyze.
> In view of this, we believe our setup is a useful first step, given that we introduce a new method.

---

> > ### Comment · Reviewer_a4f1 · 2024-11-21
> > **Response to authors**
> >
> > Thank you for clarifying my questions. I’ll keep my score.

---

### Author Response · Authors · 2024-11-21
**References**

In this thread we discuss the relation between our work and the references suggested in the reviews.
We thank the reviewers for these suggestions, all of which will be added to the paper.

Below we provide notes on all 8 references from the reviews.

We note that our original discussion of the fairness-accuracy Pareto front literature in the paper concentrated on the Pareto front of *representations*, since this is our main subject, while the references below also involve the tradeoffs for classifiers, which is a different, easier problem.


In particular, on one hand, our results with the binary error $h$ (eq. (4)) may  be used (Sections 3, 6) to derive also classification tadeoffs with this error.
However, if one uses the *entropy*  $h$ (line 178), then there is no associated classification problem at all, and all classification work is simply incomparable.


As an overview of the discussion below:
**(a)** All work below invloving  representations considers much more complex models than ours, in which guarantees would be much harder to obtain. In addition,  our results on the structure of fair representations, and the resulting algorithm, are new.

**(b)** Perhaps the approach closest in spirit to our work is the very recent work [1] on classification, where the authors start with a somewhat similar outlook on the problem, but derive a very different algorithm. This makes for an interesting comparison. Thanks again to R5 for this reference!

We have started working on adding an empirical comparison of our approach to [1]. However, unfortuantely, their code does not run out of the box. Moreover, they considerbaly modify the datasets (subsampling, manual feature selection, custom discretisation of continuous features). All these modifications have nothing to do with fairness. This makes direct comparison impossible. With these issues resolved, an empirical comparison will be included in the final version of the paper.



The following papers are discussed below:

* [1] Wang, Hao, et al. "Aleatoric and epistemic discrimination: Fundamental limits of fairness interventions." Advances in Neural Information Processing Systems 36, December 2023.

* [2] Dehdashtian, Sepehr, et al. "Utility-Fairness Trade-Offs and How to Find Them." Proceedings of the IEEE/CVF Conference on Computer Vision and Pattern Recognition, 2024.

* [3] Sadeghi, Bashir, et al. "On characterizing the trade-off in invariant representation learning." Transactions on Machine Learning Research, 2022.

* [4] Menon, Aditya Krishna, and Robert C. Williamson. "The cost of fairness in binary classification." Conference on Fairness, Accountability and Transparency, PMLR, 2018.

* [5] Zhao, Han, et al. "Inherent tradeoffs in learning fair representations." Journal of Machine Learning Research, 2022.

* [6] Kim, Joon Sik, Jiahao Chen, and Ameet Talwalkar. "FACT: A diagnostic for group fairness trade-offs." International Conference on Machine Learning (ICML) 2020.

* [7] Hamman, Faisal, and Sanghamitra Dutta. "Demystifying Local & Global Fairness Trade-offs in Federated Learning Using Partial Information Decomposition." International Conference on Learning Representations (ICLR) 2024.

* [8] Lechner et al, Impossibility results for fair representations, arxiv 2021


Continued below

---

> ### Author Response · Authors · 2024-11-21
>
> ## [1] Aleatoric and epistemic discrimination: Fundamental limits of fairness interventions.
>
>
> This paper starts with the observation
> (made earlier in [6] below) that the classification Pareto front depends only on confusion matrices produced by all possible classifiers on the data. Our factorisation result is analogous to this observation (but is also much more general).
>
> The main contribution of this paper is a way to characterise the above family of confusion matrices, called $\mathcal{C}$. Specifically,
> $\mathcal{C}$ is characterised as an intersection of an infinite number of constraints. The suggested algorithm iteratively adds constraints and solves the Pareto problem with increasingly tighter versions of $\mathcal{C}$, i.e.
> $\mathcal{C}_1 \supset \mathcal{C}_2 \ldots ... \supset \mathcal{C}$.
>
> Similarly to our approach, the method requires learning the probabilities estimator $g(x,a) = P(y|x,a)$, which is used to build the constraints.
>
>
> Our approach, MIFPO, however, uses the estimator $g$ differently. We reduce the required search space by idenitifying necessary proeprties of optimal fair representations, and then constarct a closed form problem, with a finite number of constraints, which can be solved by standard solvers, rahter than requiring custom iterative algorithms.
> We would also like to reitarate the we solve the more general problem of representations, rather than just classification problem.
>
>
> It is worth mentioning that this paper supports multi-class classification and non-binary discrete attributes. However, this means that the iterative algorithm described above operates in $|A|\cdot|C|$ dimensional space. Note also that the complexity of convex bodies (in terms of # of hyperplane constarints) may grow exponentially with dimension. Thus it is not obvious that this approach would be advatageous to the discretisaion of the silmplex $\Delta_{Y}$, which is reqiured by MIFPO, especially if such discretisation is performed with care, via clustering algorithms.
> We belive the exloration of this question is an interesting direction for future research.
>
>
>
> ## [2] Utility-Fairness Trade-Offs and How to Find Them.
>
> This is a very recent paper.
> Here a  representation learning scheme is proposed that utilizes learnable neural networek encoders, followed by RKHS (reproducing kernel Hilber space) maps and RKHS classifiers.
> Fairness is expressed via independence, which in turn is measured via a certain well known RKHS based proxy (covariance *operator*).
>
> While there are novel components in this paper, and multiple advatages, such as treatment of continuous sensitive variables, from the point of view of our work, this paper is similar to earlier work on representations, such Song et al 2019, and FNF.
>
> Specifically, it is complex, involving neural networks, and the use of RKHS introduces several implicit assumptions. In particular, the sample complexity of the RKHS proxy measure and its relation to, say, standard mutual information, are not well understood.
>
> Clearly, while our approach requires a discrete sensitive attribute (binary, in current implementation), it is much simpler, has less implicit assumtions, and sources of possible errors are clearer.
>
> We also note that it would be difficult to empirically compare this method to ours, since there is no simple relation between the natural total variation based fairness criterion that we use and the covariance operator based criterion of this paper.
>
> The advatages of using the total variation were discussed, for instance,  in "Learning adversarially fair and
> transferable representations", Madras et al 2018. See also our discussio in Section 3.
>
>
> ## [3] On characterizing the trade-off in invariant representation learning.
>
> Similarly to [2], this work also studies fair representations in RKHS setting. Notably, in the particular setting of the paper, optimal closed form solutions can be derived. On the other hand, however, the setup involves strong assumptions on the distribution of the data (Gaussianity of certain projections). Moreover, similarly to the paper [2] above, additional assumptions are also implicit in the very use itself of the RKHSs. Again, this is an approach that is fundamentally different from ours, is more complex and with much stronger assumptions.
>
>
> ## [4] The cost of fairness in binary classification.
>
> The setup studied in this paper is that of *cost-sensitive classification*.
> This setup is different from the standard clasification with accuracy or with any loss involving expectation of cost.
>
> We also refer to the discussions in  [1] (their Section 3), and in [5] (their Section 8), related to this point, and discussing this particular paper.
>
> *Continued below*

---

> ### Author Response · Authors · 2024-11-21
>
> ## [5] Inherent tradeoffs in learning fair representations.
>
> The main contributions of this paper are dedicated to the development of  certain theoretical error bounds for the *perfecly fair* classifier and are not concerned with the full Pareto front. (i.e. when we also allow gradually less fairness).
>
> Using these results, bounds for partial fairness are then derived effectively as a corollary from the perfect fairness results, using stndard argumnets.
>
>
> Similarly to the case of FACT ([6] below), such bounds depend on the data very weakly, only through 4 scalars, and result in linear bounds on Pareto front, that can be significantly improved.
> See also our discussion of [6] and [1].
>
>
> We also note that this paper does not provide algorithms for learning classifiers or representations, and only  studies abstract inequalities.
>
>
> ## [6] FACT: A diagnostic for group fairness trade-offs.
>
> This paper is a predecessor of the paper [1] above. For the classification setting, the paper developed some simple bounds which were then improved in [1] by significantly stregthening the dependence on the data.
>
> ## [7] Demystifying Local & Global Fairness Trade-offs in Federated Learning Using Partial Information Decomposition.
>
> This paper deals with the federated learning -- a situation where a model is trained by multiple parties, each with different dataset.
>
> The tradeoff condisered here is different from the one we study. Indeed, rather than the fairness-accuracy tradeoff on a single dataset, this paper studies the tradeoff  between local and global fairness (fairness on individual datasets of the participants vs on aggregate).
>
>
>
> ## [8] Lechner et al, Impossibility results for fair representations, arxiv 2021
>
> This paper cosiders some issues with Fair Representations under distribution shifts.
> Our understanding is that it does not deal directly with the accuracy-fariness tradeoff, and does not provide algorithms  for evaluating these quantities. In our setting, distribution shifts are not studied.

---

> > ### Author Response · Authors · 2024-11-26
> >
> > Dear Reviewers,
> >
> > We have incorporated many of the discussions suggested in an updated version of the paper, which is now available. To accommodate these additions while maintaining the page limit, we have relocated most of the classification-related evaluations to the supplementary material.
> >
> > ### Please note that this is not a final version, and we plan to further revise the paper to incorporate the feedback from all reviewers in the final version.

---

### Meta-Review · Area_Chair_FVVL · 2024-12-11

**Metareview:**

This paper introduces a method to compute the Pareto front for fairness-constrained learning tasks. The key step is to transform the problem into a lower-dimensional space. Focusing on binary classification, this method is very efficient because the low-dimensional space can be efficiently discretized. The reformulated problem involves concave minimization, which can be solved by convex-concave programming methods. The authors validate their method through experiments on multiple standard fairness datasets.

After reading reviewers' feedbacks and reading this paper, I recommend rejection. Here are the main concerns on the proposed appraoch.

1. This work only considers a binary sensitive variable. The authors claim they can extent it for the multi-group case. However, the size of the optimization problem increases exponentially with the number of groups (and it is noncovex). The size is $|S_0|\times |S_1|\times k$ for the binary case and will be $|S_0|\times |S_1|\cdots\times|S_C|\times k$ when there are $C$ cases. This limits the applicability of the proposed method.

2. This method only works for binary classification because the computational cost also increases exponentially with the number of classes. In the binary classification case, the simplex $\Delta_Y$ is a 1-D line segment, which can be discretized easily. If there are $P$ classes, the proposed method has to discretize a simple $\Delta_Y$ in $\mathbb{R}^P$, so the grid size will be exponential in $P$.

3. The authors claim the concave minimization can be solved by existing solver. I don't think it is always easy. It is a nonconvex problem so the problem size cannot be large. However, to get a good approximation of the distribution, they need to create a fine-grid histogram, which will not be a small-size problem. Let alone the size increases with the number of classes and the number of sensitive groups as mentioned above.

4. The author assume the classification model can exactly learn or closely approximate the true probability  $P(Y|X, a)$, which is not always true even with post-training calibration. However, a method that improves the fairness of a model should not have to assume the model's classification accuracy.

Also, the authors should be careful about using the term "representation learning" to describe their method. Typically, representation learning is to learn a representation of data that provides useful information for various downstream tasks. However, the representation in this work is simply a grid of interval $[0,1]\times [0,1]$, so such a representation is only for this specific binary classification task.

**Additional Comments On Reviewer Discussion:**

Reviewer a4f1 gave a rather short review but at least he/she pointed a limitation of this work, which I totally agree. The limitation is the proposed method only works for binary sensitive variable. I provided more details on this point in my Meta Review.

Reviewer Aqxr and Reviewer YRP9 acknowledge that they are not the expert in fairness representation learning although their reviews are relatively long. Reviewer Aqxr is somewhat influenced by Reviewer YRP9 (again, not an expert on this topic) to increase the score. Therefore, I will give lower weights on their scores.

Review mT9S and Review G4EM provided detailed feedbacks. I think Review G4EM did raise some good questions and reasonable concerns. It is just unfortunate that Review G4EM did not complete his/her discussion with the authors.

Given the situation, I have to read the paper for some details and come up with the recommendation of rejection.

---

### Decision · Program_Chairs · 2025-01-22

Reject